# CHPO: Constrained Hybrid-action Policy Optimization for Reinforcement Learning

**Ao Zhou**[1,2]    **Jiayi Guan**[1]    **Li Shen**[3]    **Fan Lu**[1]    **Sanqing Qu**[1]    **Junqiao Zhao**[1]
**Ziqiao Wang**[1]    **Ya Wu**[4]    **Guang Chen**[1,2]*

[1]Tongji University
[2]Shanghai Innovation Institute
[3]Sun Yat-Sen University
[4]CNNC Equipment Technology Research (Shanghai) Co., Ltd.
{2211107, guanjiayi, lufan, sanqingqu}@tongji.edu.cn
{Zhaojunqiao, ziqiaowang, guangchen}@tongji.edu.cn
mathshenli@gmail.com
qpo144@163.com

## Abstract

Constrained hybrid-action reinforcement learning (RL) promises to learn a safe policy within a parameterized action space, which is particularly valuable for safety-critical applications involving discrete-continuous hybrid action spaces. However, existing hybrid-action RL algorithms primarily focus on reward maximization, which faces significant challenges for tasks involving both cost constraints and hybrid action spaces. In this work, we propose a novel **C**onstrained **H**ybrid-action **P**olicy **O**ptimization algorithm (CHPO) to address the problems of constrained hybrid-action RL. Concretely, we rethink the limitations of hybrid-action RL in handling safe tasks with parameterized action spaces and reframe the objective of constrained hybrid-action RL by introducing the concept of Constrained Parameterized-action Markov Decision Process (CPMDP). Subsequently, we present a constrained hybrid-action policy optimization algorithm to confront the constrained hybrid-action problems and conduct theoretical analyses demonstrating that the CHPO converges to the optimal solution while satisfying safety constraints. Finally, extensive experiments demonstrate that the CHPO achieves competitive performance across multiple experimental tasks. Our code is available at github.CHPO.

## 1 Introduction

Hybrid-action reinforcement learning (RL) aims to handle tasks with parameterized action spaces, combining discrete and continuous actions to enable effective decision-making in complex scenarios [1–3]. Recent works of hybrid-action RL have achieved remarkable achievements in the domains of policy games [4, 5], robotics [6–9], resource allocation [10–12], and autonomous driving [13, 14]. However, safety concerns remain a primary challenge to the real-world deployment of hybrid-action RL, particularly in scenarios with high safety requirements [15–17]. For instance, in the context of autonomous driving, agents need to learn a reward-maximizing policy within parameterized action spaces while simultaneously ensuring collision avoidance [18–20]. Constrained hybrid-action RL is a promising and potentially effective approach to address the aforementioned issues. It learns policies that satisfy safety constraints in tasks with parameterized action spaces.

---

*Corresponding author

39th Conference on Neural Information Processing Systems (NeurIPS 2025).

Although substantial progress is being made by leveraging the Parameterized Action Markov Decision Process (PAMDP) framework [21] for policy learning in hybrid action tasks [2, 5, 22], its adaptation to handle safety constraints in such settings remains challenging and unresolved [23–25]. For the aforementioned cost constraint problems, common methods, like the Lagrange multiplier and penalty function methods [26–29], are sensitive to initial values and require extra coefficients, leading to training instability and costly hyperparameter tuning. Furthermore, since hybrid-action RL requires the simultaneous optimization of discrete actions and continuous parameters within policy networks, this typically necessitates the design of evaluation strategies for both components to jointly guide policy updates, which increases the difficulty of convergence compared to RL for a single action space. To make matters worse, the additional hyperparameters introduced by safety constraints further exacerbate the instability of model training. Therefore, this work focuses on solving the problems of constrained hybrid-action RL involving hybrid actions and cost constraints.

To solve the aforementioned constrained hybrid-action optimization problem in parameterized action spaces, we propose a novel **C**onstrained **H**ybrid-action **P**olicy **O**ptimization algorithm (CHPO). Concretely, we introduce the Constrained Parameterized-action Markov Decision Process (CPMDP) to represent the cost constraints in parameterized action spaces. Based on this, we redefine a novel constrained hybrid-action RL objective for tasks involving hybrid actions and cost constraints. Subsequently, we propose a constrained hybrid-action policy optimization algorithm to address the constrained hybrid-action RL tasks. Furthermore, we present theoretical analyses on the convergence and safety of CHPO, demonstrating its capability to adaptively learn safe policies for parameterized action spaces under given safety constraints. Finally, extensive experiments demonstrate that the CHPO algorithm achieves competitive performance across multiple experimental tasks, particularly outperforming baseline algorithms in maximizing rewards while ensuring that the average cost across multiple seeds satisfies the safety constraint. The main contributions of this work are listed as follows:

- To the best of my knowledge, we are the first to formulate a constrained hybrid-action RL objective for the policy optimization problem involving hybrid actions and cost constraints.

- We develop a constrained hybrid-action policy optimization method to solve the constrained hybrid-action RL tasks, which not only learns safe policies that satisfy safety constraints but also improves the stability of the algorithm.

- Theoretical analyses provide the convergence and safety guarantees of CHPO, indicating CHPO can learn safe policies for hybrid action spaces under given safety constraints.

- Extensive comparisons and ablation experiments demonstrate that the CHPO algorithm delivers competitive performance, particularly outperforming baseline algorithms in maximizing rewards while ensuring that average costs across multiple seeds satisfy safety constraints.

## 2   Related Work

In this section, we extensively discuss the related work on constrained hybrid-action RL. We primarily focus on hybrid-action RL and constrained RL. In addition, we provide a supplementary discussion of the related work on RL in Appendix A.

**Hybrid-action RL** is an approach to address parameterized actions composed of discrete actions with continuous action parameters. The straightforward method is to directly discretize the continuous action spaces and transform them into a large discrete set, which often results in an excessively large discrete action space [30]. Currently, the PAMDP [21] framework has been widely adopted to better address the policy learning challenges in RL with parameterized action spaces. Based on it, PADDPG [1] applies the DDPG [31] algorithm directly by relaxing discrete action spaces into a continuous set, and PDQN [4] combines the spirits of both DQN [32] for discrete action spaces and DDPG for continuous action spaces. Furthermore, MPDQN [5] utilizes multi-pass deep Q-networks to separate continuous action parameters, and HPPO [2] uses the hybrid actor-critic architecture flexible to the structure of the action space. Moreover, some studies focus on considering the dependency between discrete and continuous actions to solve RL tasks with hybrid actions [8, 33].

**Constrained RL** aims to solve policy optimization problems where cost constraints are enforced alongside maximizing rewards. After the proposal of the Constrained Markov Decision Process (CMDP) [34] framework, numerous constrained RL algorithms have been developed to learn safe policies satisfying cost constraints [35, 36, 16, 37]. Among them, some works directly use

Lagrange multipliers and show outstanding performance in satisfying constraints [38–41]. Editor policies and safe exploration have also achieved promising results as alternatives to Lagrangian multipliers [42, 43]. Furthermore, there have been successful attempts in addressing constrained RL tasks by employing constrained methods such as interior-point [26], conditional value-at-risk [44–46], penalty function methods [28], and world models [47, 48]. Moreover, the algorithms based on the primal-dual methods have been widely applied to solve constrained RL problems, such as PDO [49], CVPO [50], and RCPO [51]. Extensive research on offline safe RL further enhances the safety of the training process [52–57]. These diverse methodologies underline the versatility and growing sophistication of constrained RL research.

In summary, compared to purely discrete or continuous action spaces, parameterized action spaces are more challenging to handle in RL. This difficulty arises because parameterized action spaces include both discrete actions and continuous parameters, whereas most RL models are designed specifically for either discrete or continuous action spaces. More critically, the difficulty further exacerbates the challenge of selecting safe discrete actions and continuous action parameters in safety-critical parameterized action spaces when both cost and reward are involved. When extending constrained RL algorithms to hybrid action spaces naively, directly relaxing them into a continuous space substantially increases action complexity and degrades performance, making existing constrained PPO variants [29] inapplicable. To the best of our knowledge, our work is the first attempt to simultaneously address parameterized actions and cost constraints.

## 3 Preliminaries

This section presents fundamental concepts of RL, hybrid-action RL, and constrained RL. Subsequently, we rethink the existing works and highlight the purpose and significance of our work.

### 3.1 Concepts and Background

Standard RL employs the Markov Decision Process (MDP) framework defined by the tuple $(\mathcal{S}, \mathcal{A}, P, r, \rho_0, \gamma)$, where $\mathcal{S} \in \mathbb{R}^n$ is the state space, $\mathcal{A} \in \mathbb{R}^u$ is the single discrete or continuous action space, $P : \mathcal{S} \times \mathcal{A} \times \mathcal{S} \to [0, 1]$ denotes the state transition probability $p(s_{t+1}|s_t, a_t)$ from state $s_t$ to state $s_{t+1}$ under the action $a_t$, $r : \mathcal{S} \times \mathcal{A} \to \mathbb{R}$ represents the reward function abbreviated as $r_t = r(s_t, a_t)$, $\rho_0 : \mathcal{S} \to [0, 1]$ is the distribution of initial states, and $\gamma \in (0, 1]$ is the discount factor. The policy $\pi$ is a probability distribution mapping the state $s_t$ to the action $a_t$. The common objective of standard RL is to maximize the cumulative rewards:

$$\pi^* = \arg\max_{\pi} \mathbb{E}_{\tau \sim \pi} \left[ \sum_{t=0}^{\infty} \gamma^t r_t \right], \tag{1}$$

where the $\tau = \{s_0, a_0, \cdots\} \sim \pi$ denotes the trajectory. Moreover, constrained RL is based on the CMDP framework to define a tuple $(\mathcal{S}, \mathcal{A}, C, P, r, \rho_0, \gamma)$, where the added $C$ is the set of costs $\{c_i : \mathcal{S} \times \mathcal{A} \to \mathbb{R}_+, i = 1, 2, \cdots, m\}$ for violating $m$ constraints. We use shorthand $c_{i,t} = c_i(s_t, a_t)$ for simplicity. The goal of constrained RL is to maximize the cumulative rewards $\pi^* = \arg\max_{\pi} \mathbb{E}_{\tau \sim \pi}[\sum_{t=0}^{\infty} \gamma^t r_t]$ while satisfying safety constraints $\mathbb{E}_{\tau \sim \pi}[\sum_{t=0}^{\infty} \gamma^t c_{i,t}] \leq \bar{c}_i$, where $\bar{c}_i$ is the cost threshold of the $i$-th cost constraint.

On the other hand, hybrid-action RL utilizes the PAMDP to define the parameterized tuple $(\mathcal{S}, \mathcal{A}_p, P, r, \rho_0, \gamma)$. The whole parameterized action space $\mathcal{A}_p$ includes a finite set of discrete actions $\mathcal{A}_d = \{a_1, a_2, \cdots a_k\}$ and a set of real-valued continuous parameters $\mathcal{A}_c \subseteq \mathbb{R}^{\mathcal{A}_d}$ corresponding to each discrete action $a_d \in \mathcal{A}_d$. In this way, a complete action in step $t$ is composed of $a_{d,t}$ and $a_{c,t}$, where $a_{d,t} \in \mathcal{A}_d$ is a discrete action and $a_{c,t} \in \mathcal{A}_c$ is the chosen parameter to execute with the discrete action $a_{d,t}$. The whole action space $\mathcal{A}_p$ is the union of each discrete action with all possible parameters for that action:

$$\mathcal{A}_p = \bigcup \{(a_d, a_c)|a_d \in \mathcal{A}_d, a_c \in \mathcal{A}_c\}. \tag{2}$$

The parameterized policy $\pi_p(a_{d,t}, a_{c,t}|s_t)$ is to select the discrete action $a_{d,t}$ and the continuous parameter $a_{c,t}$, which are unified as $a_t = (a_{d,t}, a_{c,t})$. Similar to standard RL, the objective of hybrid-action RL is also to maximize rewards:

$$\pi_p^* = \arg\max_{\pi_p} \mathbb{E}_{\tau \sim \pi_p} \left[ \sum_{t=0}^{\infty} \gamma^t r_t \right]. \tag{3}$$

## 3.2 Rethinking to Hybrid-action RL

We rethink the application requirements of hybrid-action RL algorithms in real-world scenarios. In the autonomous driving parking task [58, 59], the intelligent vehicle needs to select the direction of travel and the distance to travel in that direction to successfully park, but the vehicle must avoid collisions with obstacles ensuring safety throughout the process. In this work, we establish and test a parameterized *Parking* task with specific details provided in Appendix D.1. Similarly, in robotic manipulation, both the selection of operations and the corresponding operation parameters must be considered, along with constraints related to hazardous areas. These cases require us to simultaneously account for parameterized actions and cost constraints.

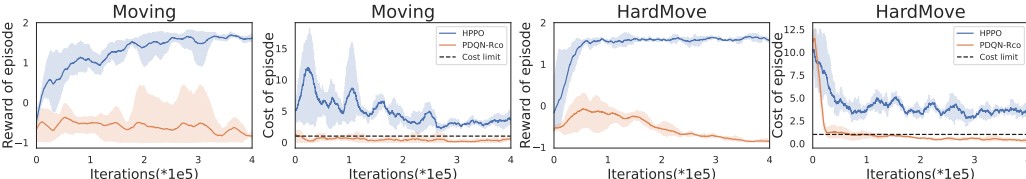

Figure 1: The figure depicts the reward and cost curves for the *Moving* and *HardMove* tasks. The shaded curves represent the mean and variance of online testing during the training process with three different random seeds. The cost limit is set as $\bar{c}_i = 1$.

We conduct experiments on the hybrid-action RL algorithms of PAMDP in the typical experimental scenarios *Moving* [2, 22, 33, 60] and *HardMove* [22, 33, 60] of DI-engine [60] where we introduce costs and hazardous areas into these tasks. A detailed description of the tasks is provided in Appendix D.1. The experimental results are shown in Fig. 1. The curve of the HPPO [2] algorithm indicates that, while the current hybrid-action RL algorithm can achieve high rewards, it often violates safety constraints to a significant extent. Additionally, inspired by the RCPO [51] algorithm, we attempt to use the cost as a negative reward to learn policies that satisfy safety constraints. We combine the PDQN algorithm [4] with reward-constrained policy optimization to form PDQN-Rco. The result of PDQN-Rco shows that although the reward-constrained method can learn policies that adhere to safety constraints, the rewards are relatively poor. This suggests that the reward-constrained method for hybrid action spaces tends to drive the policy towards local optima, making it challenging to ensure optimal performance.

In summary, for parameterized action spaces, the current hybrid-action RL algorithms and the reward-constrained approaches are insufficient to adequately meet the requirement of maximizing rewards while satisfying safety constraints. Inspired by CMDP and PAMDP, we propose a novel constrained hybrid-action RL task for addressing the simultaneous treatment of parameterized action spaces and cost constraints in RL application scenarios.

## 4 Methodology

In this section, we provide a detailed exposition of constrained hybrid-action policy optimization for constrained hybrid-action RL. Firstly, we define the constrained hybrid-action RL task and reframe the objective of constrained hybrid-action RL by introducing the concept of CPMDP. Subsequently, we construct a constrained hybrid-action actor-critic architecture to address scenarios involving costs and hybrid actions and propose a constrained hybrid-action policy optimization algorithm for constrained hybrid-action RL within the architecture. Additionally, we present theoretical analyses demonstrating that CHPO guarantees the convergence of learning safe policies under safety constraints.

### 4.1 Constrained Hybrid-action RL

**Definition 4.1.** *Constrained hybrid-action RL tasks refer to tasks that involve parameterized action spaces and cost constraints, with the objective of maximizing rewards while satisfying safety constraints. This is formulated as:*

$$\pi_p^* = \arg\max_{\pi_p} \mathbb{E}_{\tau \sim \pi_p} \left[ \sum_{t=0}^{\infty} \gamma^t r_t \right], \text{ s.t. } \mathbb{E}_{\tau \sim \pi_p} \left[ \sum_{t=0}^{\infty} \gamma^t c_{i,t} \right] \leq \bar{c}_i. \tag{4}$$

Based on the analysis and discussion in Section 3.2, it is known that existing hybrid-action RL algorithms and the reward-constrained method face challenges when dealing with tasks that involve cost constraints and parameterized actions. Therefore, we introduce the cost to define the constrained parameterized tuple $(\mathcal{S}, \mathcal{A}_p, C, P, r, \rho_0, \gamma)$ as the CPMDP framework, where the added $C$ is the set of costs $\{c_i : \mathcal{S} \times \mathcal{A}_p \rightarrow \mathbb{R}_+, i = 1, 2, \cdots, m\}$ for violating $m$ constraints in parameterized action spaces and we also use shorthand $c_{i,t} = c_i(s_t, a_t) = c_i(s_t, (a_{d,t}, a_{c,t}))$ for simplicity.

## 4.2 Constrained Hybrid-action Policy Optimization

Due to the difficulty in directly handling the objective stated in Definition 4.1, we are inspired by HPPO [2] to define the state-value for reward on the parameterized policy $\pi_p$ as $V^r(s) = \mathbb{E}_{\tau \sim \pi_p}[\sum_{t=0}^{\infty} \gamma^t r(s_t, a_t)|s_0 = s]$ and the state-value for cost is defined similarly as $V^{c_i}(s) = \mathbb{E}_{\tau \sim \pi_p}[\sum_{t=0}^{\infty} \gamma^t c_{i,t}(s_t, a_t)|s_0 = s]$. Furthermore, since the policy $\pi_p$ outputs two types of actions, we define the discrete policy $\pi_d$ outputting discrete actions $a_d$ and the continuous policy $\pi_c$ outputting continuous parameters $a_c$ separately, which together form the policy $\pi_p$ for a clearer explanation. Therefore, the Eq. (4) is rewritten as:

$$\pi_{\theta_p}^* = \arg\max_{\pi_{\theta_p}} \mathbb{E}_{s \sim \mathcal{S}}\big[V_{\phi_r}^r(s)\big], \text{ s.t. } \mathbb{E}_{s \sim \mathcal{S}}\big[V_{\phi_{c_i}}^{c_i}(s)\big] \leq \bar{c}_i, \tag{5}$$

where $\phi_r$ and $\phi_{c_i}$ represent the parameters of the state-value functions, and $\theta_p = (\theta_d, \theta_c)$ is the parameters of the policy $\pi_p$ consisting of $\theta_d$ and $\theta_c$.

To handle a class of discrete-continuous hybrid action spaces with safety constraints, we propose a constrained hybrid-action actor-critic architecture capable of learning safe policies in both discrete and continuous action spaces under cost constraints. The architecture comprises two critic networks and two actor networks: the reward critic network represents the reward state-value function $V_{\phi_r}^r(s)$, the cost critic network represents the cost state-value function $V_{\phi_{c_i}}^{c_i}(s)$, the discrete actor network defines the discrete policy function $\pi_{\theta_d}$, and the continuous actor network defines the continuous policy function $\pi_{\theta_c}$. In our architecture, the cost critic network provides the estimation of the cost advantage function $\hat{A}_t^{c_i} = -V^{c_i}(s_t) + c_{i,t} + \gamma c_{i,t+1} + \cdots + \gamma^{T-t-1} c_{i,T-1} + \gamma^{T-t} V^{c_i}(s_T)$ to allow the constrained hybrid-action actor-critic architecture to flexibly adapt to constrained policy optimization methods, where $T$ is timesteps for the policy $\pi_p$ and much less than the length of an episode, and $t \in [0, T]$ is the timestep index. The computation of the reward advantage function $\hat{A}^r$, similar to $\hat{A}^{c_i}$, is given by $\hat{A}_t^r = -V^r(s_t) + r_t + \gamma r_{t+1} + \cdots + \gamma^{T-t-1} r_{T-1} + \gamma^{T-t} V^r(s_T)$. The objective of the cost state-value update is written as:

$$\mathcal{L}(\phi_{c_i}) = \arg\min_{\phi_{c_i}} \mathbb{E}_{s \sim \mathcal{S}}\Big[\big(\hat{C}_i(s) - V_{\phi_{c_i}}^{c_i}(s)\big)^2\Big], \tag{6}$$

where $\hat{C}_i(s) = \sum_{j=0}^{\infty} \gamma^j c_{i,t+j}(s_{t+j}, a_{t+j})$ is the target cost return computed by the Monte Carlo method and $s_t = s$. Similarly, the objective of the reward state-value update is expressed as:

$$\mathcal{L}(\phi_r) = \arg\min_{\phi_r} \mathbb{E}_{s \sim \mathcal{S}}\Big[\big(\hat{R}(s) - V_{\phi_r}^r(s)\big)^2\Big], \tag{7}$$

where $\hat{R}(s)$ is the target reward return computed using $\hat{R}(s) = \sum_{j=0}^{\infty} \gamma^j r_{t+j}(s_{t+j}, a_{t+j})$ and $s_t = s$.

The constrained hybrid-action optimization described in Eq. (5) is conventionally tackled using the Lagrange multiplier method, which converts the constrained formulation into an unconstrained optimization problem by incorporating penalty terms into the objective function. While effective in principle, this primal-dual approach often suffers from issues such as sensitivity to the initialization of Lagrange multipliers and the careful selection of learning rates, making it computationally expensive due to the extensive hyperparameter tuning required. Inspired by CRPO [61], we directly tackle the constrained hybrid-action problem using the policy gradient of the primal objective based on stochastic approximation theory, effectively avoiding the instability and inefficiency associated with auxiliary Lagrange multipliers. Theoretical analyses in Section 4.3 indicate that the direct optimization method ensures stable convergence and enhances the practicality of solving constrained hybrid-action RL problems.

Based on the above analysis, we propose a **C**onstrained **H**ybrid-action **P**olicy **O**ptimization (CHPO) for RL with parameterized action spaces to solve constrained hybrid-action RL tasks. The parameterized policy update for the CHPO algorithm is performed within the constrained hybrid-action

actor-critic architecture. We first evaluate whether the current policy satisfies the cost constraints based on the cost state-value and then update the parameterized policy by maximizing the reward or minimizing the cost. Specifically, when the cost constraints in parameterized action spaces are satisfied (i.e., $\mathbb{E}_{s \sim \mathcal{S}}\left[V_{\phi_{c_i}}^{c_i}(s)\right] \leq \bar{c}_i$), the constrained hybrid-action optimization described in Eq. (5) is transformed into an unconstrained optimization problem, requiring consideration of reward maximization within the parameterized action spaces. If the cost constraints are not satisfied, the constrained hybrid-action optimization in Eq. (5) is converted into minimizing the cost in discrete-continuous hybrid action spaces. Therefore, the objective of the policy update is expressed as:

$$\mathcal{L}(\theta_p) = \arg\max_{\theta_p} \mathbb{E}_{s \sim \mathcal{S}} \left[ \frac{\pi_{\theta_p}(a|s)}{\pi_{\theta_p^k}(a|s)} \cdot \left( \mathbb{I}_{\pi_{\theta_p} \in \pi_s} \hat{A}^r - \mathbb{I}_{\pi_{\theta_p} \notin \pi_s} \hat{A}^{c_i} \right) \right], \tag{8}$$

where $\pi_s$ represents the safe policy, $\pi_{\theta_p} \in \pi_s$ indicates that the policy $\pi_{\theta_p}$ satisfies safety constraints, while $\pi_{\theta_p} \notin \pi_s$ denotes that the policy $\pi_{\theta_p}$ violates the constraints. $\mathbb{I}_{\pi_{\theta_p} \in \pi_s}$ is a sign function that takes the value one when $\pi_{\theta_p}$ satisfies safety constraints; otherwise, it takes zero. Similarly, when $\pi_{\theta_p}$ violates safety constraints, $\mathbb{I}_{\pi_{\theta_p} \notin \pi_s}$ is equal to one; otherwise, it equals zero. $\theta_p^k$ is the parameters of the discrete and continuous actor networks at $k$-th update including $\theta_d^k$ and $\theta_c^k$. The policy $\pi_{\theta_p}$ update involves separately updating the discrete policy $\pi_{\theta_d}$ and the continuous policy $\pi_{\theta_c}$, which includes handling $\theta_d$ and $\theta_c$. A detailed update example is provided in Section 4.4.

### 4.3 Theoretical Analysis

To validate the convergence properties and cost safety of CHPO, we derive the policy and cost boundary. Before deriving the following, we review the policy boundary of the natural stochastic gradient, as presented in Lemma 4.2.

**Lemma 4.2.** *The boundary of the policy based on natural stochastic gradient updates [61]:*

$$\mathcal{L}(\pi_{\theta_p}^*) - \mathcal{L}(\pi_{\theta_p^k}) \leq \frac{3(1 + \alpha r_{\max})}{(1-\gamma)^2} \|Q_k^r - \hat{Q}_k^r\|_2 + \frac{2\alpha r_{\max}^2 |\mathcal{S}||\mathcal{A}|}{(1-\gamma)^3} + \frac{1}{\alpha} \mathbb{E}_{s \sim S}\left( D_{KL}(\pi_{\theta_p}^* || \pi_{\theta_p^k}) - D_{KL}(\pi_{\theta_p}^* || \pi_{\theta_p^{k+1}}) \right), \tag{9}$$

*where $r_{\max}$ represents the maximum reward, $Q_k$ and $\hat{Q}_k$ denote the evaluated value and the estimated value of the action-state at $k$-th update, $\pi_{\theta_p^k}$ and $\pi_{\theta_p^{k+1}}$ represent the policies at step $k$ and $k + 1$, respectively. The parameter $\alpha = (1-\gamma)^{1.5}/\sqrt{|\mathcal{S}||\mathcal{A}|K}$. $|\mathcal{S}|$ and $|\mathcal{A}|$ represent parameters associated with the dimensions of the observation state $s$ and the action space $a$, respectively. $K$ denotes the maximum number of policy update steps.*

Based on the policy boundary presented in Lemma 4.2 and the update method of the CHPO, we derive the policy and cost boundary as shown in Propositions 4.3 and 4.4. The propositions 4.3 and 4.4 are presented and discussed in detail in Appendices B.2 and B.3.

**Proposition 4.3.** *After updating the policy for the $K$ steps according to the method described in Eq. (8), the policy converges to a boundary.*

$$\mathcal{L}(\pi_{\theta_p}^*) - \mathbb{E}[\mathcal{L}(\pi_{\theta_p^k})] \leq \Theta\left( \frac{\sqrt{|\mathcal{S}||\mathcal{A}|}}{(1-\gamma)^{1.5}\sqrt{K}} \right). \tag{10}$$

**Proposition 4.4.** *After updating the policy for $K$ steps according to the method described in Eq. (8), the policy cost converges to a boundary.*

$$\mathbb{E}[\mathcal{L}(\pi_{\theta_p^k})] - \bar{c}_i \leq \Theta\left( \sqrt{\frac{(1-\gamma)|\mathcal{S}||\mathcal{A}|}{K}} \right). \tag{11}$$

The proposition 4.3 shows that the policy converges to the optimal policy after $K$ iterations, with the convergence bound dependent on the dimensions of the state space $|\mathcal{S}|$, the action space $|\mathcal{A}|$, and the number of update steps $K$. The proposition 4.4 demonstrates that, after $K$ iterations, the policy

cost converges to within a bounded range of the cost threshold, where the bound depends on the dimensions of the state space $|\mathcal{S}|$, the action space $|\mathcal{A}|$, and the number of iterations $K$. The number of iterations $K$ consists of the number of iterations $|\mathcal{N}_r|$ for maximizing the reward and the number of iterations $|\mathcal{N}_{c_i}|$ for minimizing the cost. We define C/A ratio$= \frac{\sum_i^m |\mathcal{N}_{c_i}|}{\sum_i^m |\mathcal{N}_{c_i}| + |\mathcal{N}_r|}$ as the proportion of update steps dedicated to cost minimization relative to the total number of policy.

**Remark 4.5.** *The proportion (C/A ratio) between the update iterations allocated to reward maximization and cost minimization influences the final optimal policy* $\mathbb{E}[\mathcal{L}(\pi_{\theta_p^k})]$.

A detailed discussion of Remark 4.5 is presented in Appendix B.4, with corresponding experiments further supplemented in the ablation study.

## 4.4 Practical Algorithm

To facilitate the understanding of the implementation process of the CHPO algorithm, we provide a detailed explanation of a practical instance of the CHPO algorithm. The pseudo-code for the CHPO algorithm is shown in **Algorithm 1** of **Appendix C**. In the state-value estimation step, we employ Eq. (6) and Eq. (7) in the constrained hybrid-action actor-critic architecture to separately update the state-values of reward and cost. In the policy update step, we evaluate whether the cost constraints are satisfied and update the policy network parameters $\theta_p$ including $\theta_d$ and $\theta_c$. Since the cost state-value function $V_{\phi_{c_i}}^{c_i}(s)$ estimates the cost return with the discount factor $\gamma$, utilizing the cost state-value in Eq. (5) to determine whether the safety constraints are met may lead to the neglect of some unsafe circumstances. Therefore, we use the cost return without the discount factor $\bar{\mathcal{L}}_{c_i}(\pi_{\theta_p^k}) = \mathbb{E}_{\tau \sim \pi_{\theta_p^k}}[\sum_{t=0}^{\infty} c_{i,t}]$ instead of $V_{\phi_{c_i}}^{c_i}(s)$ in Eq. (5) to decide whether cost constraints are satisfied and the objective of policy update, where $\pi_{\theta_p^k}$ represents the parameterized policy $\pi_{\theta_p}$ at $k$-th update. Concretely, for the parameterized policy $\pi_{\theta_p}$ update including the discrete policy $\pi_{\theta_d}$ and continuous policy $\pi_{\theta_c}$ update, if cost constraints are satisfied, the objective of policy update in Eq. (8) is transformed into:

$$\mathcal{L}(\theta_p) = \arg\max_{\theta_p} \mathbb{E}_{s \sim \mathcal{S}} \left[ \min\left( \frac{\pi_{\theta_p}(a|s)}{\pi_{\theta_p^k}(a|s)} \hat{A}^r, \text{clip}(\frac{\pi_{\theta_p}(a|s)}{\pi_{\theta_p^k}(a|s)}, 1-\epsilon, 1+\epsilon)\hat{A}^r \right) \right], \qquad (12)$$

where $\theta_p^k$ is the parameters of the parameterized policy network at $k$-th update, and $a$ includes the discrete action $a_d$ and the continuous action $a_c$. Conversely, if the safety constraints are not satisfied, the objective of the policy update is expressed as:

$$\mathcal{L}(\theta_p) = \arg\min_{\theta_p} \mathbb{E}_{s \sim \mathcal{S}} \left[ \min\left( \frac{\pi_{\theta_p}(a|s)}{\pi_{\theta_p^k}(a|s)} \hat{A}^{c_i}, \text{clip}(\frac{\pi_{\theta_p}(a|s)}{\pi_{\theta_p^k}(a|s)}, 1-\epsilon, 1+\epsilon)\hat{A}^{c_i} \right) \right]. \qquad (13)$$

That is to say, the discrete policy $\pi_{\theta_d}$ and continuous policy $\pi_{\theta_c}$ constitute the parameterized policy $\pi_{\theta_p}$ to determine complete actions, achieving reward maximization while satisfying safety constraints.

# 5 Experimental Evaluation

In this section, we conduct comprehensive comparative experiments between CHPO and previous hybrid-action RL methods in tasks with different hybrid action spaces and observation dimensions.

## 5.1 Task and Baseline

**Task.** To assess the performance of CHPO in various tasks with parameterized action spaces, we select three widely adopted tasks from DI-engine [60] and establish a *Parking* task with parameterized action spaces as experimental tasks in this work. Concretely, we choose the *Moving* [2, 22, 33, 60], *Sliding* [2, 60], and *HardMove* [22, 33, 60] tasks, all of which require agents to perform both discrete and continuous actions to reach a target area. For instance, in the *Moving* task, the agent can take discrete actions such as turn, accel, or break, accompanied by two continuous parameters: acceleration and steering angle. To the best of my knowledge, there are currently no standardized tasks for constrained hybrid-action RL. To facilitate further research and reproducibility of this work, we modify the aforementioned standard experimental scenarios of DI-engine for hybrid-action RL

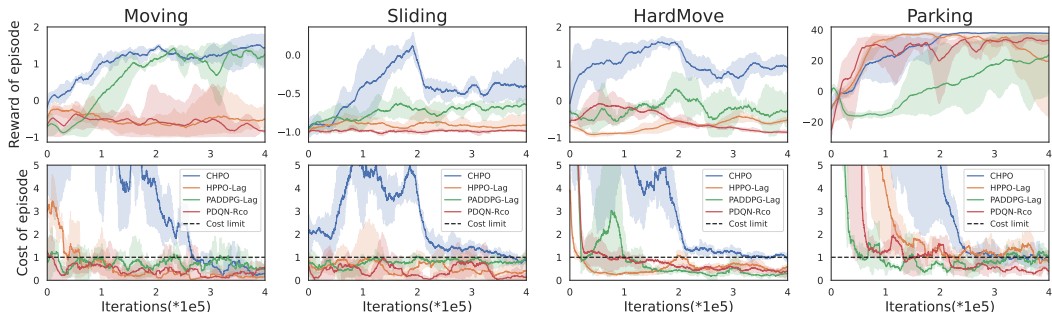

Figure 2: The figure depicts the reward and cost curves of CHPO and the baseline algorithms in the comparison experiment to showcase the performance of each algorithm. The shaded areas on the curves represent the variance obtained from online testing conducted with three random seeds. The cost limit of the four tasks is set at $\bar{c}_i = 1$.

by introducing costs and hazardous areas into these tasks. Additionally, in the *Parking* task, the intelligent vehicle selects from three discrete directional actions, each associated with a specific movement distance, to park in designated areas while avoiding collisions with obstacles and other vehicles. A more detailed description of four task settings can be found in Appendix D.1.

**Baselines.** Due to the lack of hybrid-action RL algorithms that can simultaneously handle parameterized action spaces and cost constraints, we enhance existing hybrid-action RL methods to develop constrained hybrid-action RL algorithms as the experimental baselines. PADDPG-Lag is an improved hybrid-action RL algorithm that combines the Lagrange multiplier method with PADDPG [1] to enable the optimization of constrained hybrid-action policies in parameterized action spaces. That is to say, the online safe RL algorithm DDPG-Lag handles these tasks by relaxing discrete action spaces into a continuous set. Similarly, we introduce the Lagrange multiplier method based on HPPO [2] to relax the constrained hybrid-action problem into an unconstrained optimization problem and obtain an improved algorithm HPPO-Lag. Moreover, we incorporate the reward-constrained method [51] into PDQN [4] which combines the spirits of both DQN for discrete action spaces and DDPG for continuous action spaces, resulting in an enhanced algorithm PDQN-Rco.

## 5.2 Performance Comparison Experiment

**Performance on various tasks.** To evaluate the performance of the CHPO algorithm across various safe tasks with parameterized action spaces, we conduct comprehensive comparative experiments. The results involve comparing the performance of the CHPO algorithm with the baseline algorithms in the four tasks: *Moving*, *Sliding*, *HardMove*, and *Parking*. Fig. 2 illustrates the reward and cost curves for the baseline algorithms and the CHPO algorithm in the four tasks. From the illustrated results, it can be observed that compared to the baseline algorithms, CHPO maintains the average costs within the specified cost limit while achieving higher rewards. Furthermore, the CHPO algorithm demonstrates a more stable performance compared to other baseline algorithms. Notably, CHPO provides the highest rewards across all four tasks while adhering to the cost limit. The above results indicate that CHPO consistently enforces policy compliance with safety constraints across various safe tasks involving parameterized action spaces while delivering competitive reward returns.

**Performance on different cost limits.** To evaluate the performance of the CHPO algorithm under different cost limits, we conduct comparative experiments between the CHPO algorithm and the baseline algorithms at multiple cost limits. The results in Table 1 demonstrate that the CHPO algorithm ensures adherence to cost constraints while providing competitive reward returns across the four tasks with varying cost limits. Additionally, from the results in the table, it is evident that an increase in the cost limit and more relaxed safety constraints lead to higher mean values of reward for the CHPO algorithm and the baseline algorithms in the four tasks. Notably, compared to other baseline algorithms, the CHPO algorithm effectively adjusts the policy based on different cost limits to maximize the reward as much as possible. Moreover, the reward and cost curves for the CHPO algorithm and the baseline algorithms under these cost limits $\bar{c}_i = 1.5$ and $\bar{c}_i = 2$ during online testing are provided in Fig. 4 and Fig. 5 of Appendix D.2. The analysis of the above results indicates that the CHPO algorithm can effectively learn policies that satisfy various safety constraints across tasks with parameterized action spaces while achieving excellent rewards.

Table 1: The performance of CHPO and the baseline algorithms is evaluated across different cost limits. The results from experiments involving 40 episodes are conducted with 3 random seeds. These cost limits of the four tasks are set at $\bar{c}_i = 1$, $\bar{c}_i = 1.5$ and $\bar{c}_i = 2$. The costs satisfying safety constraints are shaded gray, with the highest-reward entries highlighted in bold.

| Method | Metrics | Moving | | | Sliding | | |
|---|---|---|---|---|---|---|---|
| | | $\bar{c}_i = 1$ | $\bar{c}_i = 1.5$ | $\bar{c}_i = 2$ | $\bar{c}_i = 1$ | $\bar{c}_i = 1.5$ | $\bar{c}_i = 2$ |
| HPPO-Lag | Reward ↑ | -0.43±0.55 | -0.07±0.89 | 0.89±0.86 | -0.93±0.16 | -0.92±0.18 | -0.67±0.30 |
| | Cost ↓ | 0.17±6.58 | 0.43±8.22 | 1.09±1.41 | 0.28±1.12 | 1.04±3.95 | 1.32±3.61 |
| PADDPG-Lag | Reward ↑ | 1.30±0.67 | 1.41±0.54 | 1.50±0.52 | -0.67±0.33 | -0.61±0.32 | -0.46±0.33 |
| | Cost ↓ | 0.36±1.59 | 1.10±3.57 | 1.59±6.95 | 1.09±1.68 | 1.17±1.86 | 1.29±1.96 |
| PDQN-Rco | Reward ↑ | -0.83±0.25 | 0.22±1.02 | 0.58±0.92 | -0.99±0.09 | -0.93±0.22 | -0.89±0.39 |
| | Cost ↓ | 0.31±0.62 | 0.63±5.20 | 0.39±3.24 | 0.12±1.09 | 1.22±2.02 | 1.74±5.69 |
| CHPO | Reward ↑ | **1.41±0.55** | **1.52±0.46** | **1.60±0.36** | **-0.37±0.52** | **0.34±0.43** | **-0.30±0.51** |
| | Cost ↓ | **0.22±0.99** | **1.42±4.03** | **1.85±2.11** | **0.92±1.60** | **1.24±1.67** | **1.93±3.45** |

| Method | Metrics | HardMove | | | Parking | | |
|---|---|---|---|---|---|---|---|
| | | $\bar{c}_i = 1$ | $\bar{c}_i = 1.5$ | $\bar{c}_i = 2$ | $\bar{c}_i = 1$ | $\bar{c}_i = 1.5$ | $\bar{c}_i = 2$ |
| HPPO-Lag | Reward ↑ | -0.51±0.39 | 0.02±0.64 | 0.21±0.65 | 20.01±25.21 | 23.22±13.03 | 23.02±16.92 |
| | Cost ↓ | 0.67±1.22 | 1.08±1.95 | 1.33±1.53 | 0.82±1.26 | 1.26±3.72 | 1.75±5.24 |
| PADDPG-Lag | Reward ↑ | -0.29±0.59 | 0.80±1.20 | 1.28±0.81 | 26.66±16.39 | 35.21±5.52 | 37.24±3.00 |
| | Cost ↓ | 0.35±1.11 | 1.18±1.76 | 1.73±3.39 | 0.94±2.15 | 1.21±2.62 | 2.27±2.73 |
| PDQN-Rco | Reward ↑ | -0.88±0.20 | -0.84±0.23 | -0.81±0.24 | 32.72±7.44 | 34.95±5.30 | 35.33±5.34 |
| | Cost ↓ | 0.37±0.89 | 0.38±1.04 | 0.43±1.10 | 0.21±0.35 | 0.34±0.75 | 0.87±2.30 |
| CHPO | Reward ↑ | **0.89±0.59** | **1.02±0.54** | **1.36±0.53** | **37.63±3.04** | **37.98±2.47** | **37.73±3.15** |
| | Cost ↓ | **0.97±1.41** | **1.48±2.13** | **1.73±2.14** | **0.93±3.17** | **1.39±5.31** | **1.88±2.27** |

## 5.3 Ablation Experiment

**Performance with and without the constraint module.** We evaluate the impact of the constraint module in the CHPO algorithm by removing it during policy updates. The HPO represents CHPO without the constraint module during policy updates. Fig. 3 displays the reward and cost curves for the CHPO algorithm and the CHPO algorithm without the constraint module in the *Moving* and *HardMove* tasks. From the figure, it can be observed that when the constraint module is not used to handle safety tasks with parameterized action spaces, HPO achieves higher rewards compared to CHPO. However, the costs achieved by HPO significantly exceed the predefined cost limit $\bar{c}_i = 1$, resulting in substantial violations of safety constraints. Meanwhile, CHPO effectively adheres to the safety constraints with only a minimal loss in rewards. Additionally, the results of this ablation experiments for the *Sliding* and *Parking* tasks are presented in Fig. 6 of Appendix D.2. This demonstrates that the constraint module in the CHPO algorithm effectively enforces safety constraints in tasks with hybrid action spaces while maintaining satisfactory reward performance.

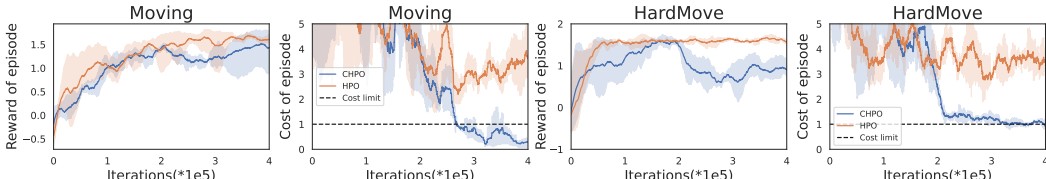

Figure 3: The figure depicts the reward and cost curves in ablation experiments regarding the constraint module. The HPO represents CHPO without the constraint module. The shadowed areas represent the variance of the test results for three random seeds and the cost limit is set at $\bar{c}_i = 1$.

**Performance on different C/A ratios.** To evaluate the impact of the C/A ratios on the performance of CHPO, we experiment with four different C/A ratios tailored to the practical training scenarios of four distinct tasks. **Fig. 7** of **Appendix D.2** presents the mean and standard deviation of CHPO's performance on safety tasks with parameterized action spaces across different C/A ratios. The results in the figure indicate that as the C/A ratios increase which reflects a higher frequency of cost minimization policy updates, the mean costs generally follow a decreasing trend. However, when the number of cost minimization policy updates becomes excessively high, the mean costs no longer decrease and instead begin to rise, accompanied by a reduction in rewards. This effect is particularly pronounced in the *Moving*, *Sliding*, and *Parking* tasks, where excessive cost minimization

policy updates result in higher cost values, lower mean rewards, and increased reward variance. Additionally, the reward and cost curves corresponding to these observations are presented in Fig. 8 of Appendix D.2. The above results indicate that the CHPO algorithm can achieve satisfactory rewards while ensuring that average costs satisfy safety constraints within a broad range of C/A ratios.

## 6  Conclusion

In this work, we propose a novel constrained hybrid-action policy optimization algorithm for constrained hybrid-action reinforcement learning. Concretely, we first rethink the requirements of hybrid-action RL in real-world applications and redefine the constrained hybrid-action RL objective for tasks involving hybrid actions and cost constraints by introducing the CPMDP. Subsequently, we present a constrained hybrid-action policy optimization algorithm to address the constrained hybrid-action RL tasks within the constrained hybrid-action actor-critic architecture. Additionally, we present theoretical analyses demonstrating that our method guarantees the convergence of learning safe policies for parameterized action spaces under given safety constraints. Finally, extensive experiments illustrate that the CHPO algorithm is capable of handling the constrained hybrid-action RL tasks and providing competitive performance, particularly in terms of safety.

## Acknowledgments and Disclosure of Funding

This work was supported by the National Key Research and Development Program of China (No. 2024YFE0211000), in part by the National Natural Science Foundation of China (No. 62372329, No. 62576364), in part by the Shanghai Scientific Innovation Foundation (No. 23DZ1203400), in part by the China Postdoctoral Science Foundation (No. BX20250383, GZB20250385, 2025M771530, 2025M771539), in part by Tongji-Qomolo Autonomous Driving Commercial Vehicle Joint Lab Project, and in part by Xiaomi Young Talents Program.

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

# A    Related Work

In this section, we further discuss the related work on standard RL algorithms to illustrate the challenges in handling hybrid action spaces.

**RL** seeks to learn reward-maximizing policies through interactions with environments characterized by a single action space. On the one hand, RL for discrete action spaces has been extensively studied, focusing on tasks where the set of possible actions is finite. Based on the Q-learning [62] algorithm which represents the expected cumulative reward of taking action, some variations of DQN [63] are widely used in discrete action spaces, including asynchronous DQN [64], double DQN [65], and dueling DQN [32]. On the other hand, policy gradient [66] methods, such as PPO [67] and DDPG [31], have shown effectiveness in handling continuous action spaces by directly optimizing policies using a gradient. Extensions like TRPO [68] and SAC [69] improve stability and efficiency through constraints on policy updates and entropy regularization. Since most RL models are specifically designed for either discrete or continuous action spaces, they struggle to simultaneously handle action spaces that contain both discrete and continuous components.

# B    Proofs and Discussions

**Lemma B.1.** *(Policy gap [61]) The policy gap for the natural stochastic gradient updated policy is:*

$$
\mathcal{L}(\pi_{\theta_p}^*) - \mathcal{L}(\pi_{\theta_p^k}) \leq \frac{3(1+\alpha r_{\max})}{(1-\gamma)^2}\|Q_k^r - \hat{Q}_k^r\|_2 + \frac{2\alpha r_{\max}^2|\mathcal{S}||\mathcal{A}|}{(1-\gamma)^3} + \frac{1}{\alpha}\mathbb{E}_{s\sim S}\big(D_{KL}(\pi_{\theta_p}^*||\pi_{\theta_p^k}) \\ - D_{KL}(\pi_{\theta_p}^*||\pi_{\theta_p^{k+1}})\big),
\tag{14}
$$

*where $r_{\max}$ represents the maximum reward, $Q_k$ and $\hat{Q}_k$ denote the evaluated value and the estimated value of the action-state at $k$-th update, $\pi_{\theta_p^k}$ and $\pi_{\theta_p^{k+1}}$ represent the policies at step $k$ and $k+1$, respectively. The parameter $\alpha = (1-\gamma)^{1.5}/\sqrt{|\mathcal{S}||\mathcal{A}|K}$.*

**Proposition B.2.** *After updating the policy for the $K$ steps according to the method described in Eq. (8), the policy converges to a boundary.*

$$
\mathcal{L}(\pi_{\theta_p}^*) - \mathbb{E}[\mathcal{L}(\pi_{\theta_p^k})] \leq \Theta\left(\frac{\sqrt{|\mathcal{S}||\mathcal{A}|}}{(1-\gamma)^{1.5}\sqrt{K}}\right).
\tag{15}
$$

***Proof.*** The relationship between the policy after $K$ steps of updates and the optimal policy is represented as $\mathcal{L}(\pi_{\theta_p}^*) - \mathcal{L}(\pi_{\theta_p^k})$.

$$
\mathcal{L}(\pi_{\theta_p}^*) - \mathbb{E}[\mathcal{L}(\pi_{\theta_p^k})]
$$

$$
= \frac{1}{\sum_{i=0}^m |\mathcal{N}_{c_i}| + |\mathcal{N}_r|}\left[\left(\sum_{i=0}^m \sum_{k\in\mathcal{N}_{c_i}} + \sum_{k\in\mathcal{N}_r}\right)\mathcal{L}(\pi_{\theta_p}^*) - \sum_{i=0}^m\sum_{k\in\mathcal{N}_{c_i}}\mathcal{L}_{c_i}(\pi_{\theta_p^k}) - \sum_{k\in\mathcal{N}_r}\mathcal{L}_r(\pi_{\theta_p^k})\right]
$$

$$
= \frac{\sum_{i=0}^m\sum_{k\in\mathcal{N}_{c_i}}}{\sum_{i=0}^m|\mathcal{N}_{c_i}|+|\mathcal{N}_r|}\left[\mathcal{L}(\pi_{\theta_p}^*) - \mathcal{L}_{c_i}(\pi_{\theta_p^k})\right] + \frac{\sum_{k\in\mathcal{N}_r}}{\sum_{i=0}^m|\mathcal{N}_{c_i}|+|\mathcal{N}_r|}\left[\mathcal{L}(\pi_{\theta_p}^*) - \mathcal{L}_r(\pi_{\theta_p^k})\right]
$$

$$
\overset{\{i\}}{\leq} \frac{\sum_{i=0}^m\sum_{k\in\mathcal{N}_{c_i}}}{\sum_{i=0}^m|\mathcal{N}_{c_i}|+|\mathcal{N}_r|}\left[\frac{1}{\alpha}\mathbb{E}_{s\sim S}\big(D_{KL}(\pi_{\theta_p}^*||\pi_{\theta_p^k}) - D_{KL}(\pi_{\theta_p}^*||\pi_{\theta_p^{k+1}})\big) + \frac{2\alpha c_{\max}^2|\mathcal{S}||\mathcal{A}|}{(1-\gamma)^3}\right.
$$

$$
\left. + \frac{3(1+\alpha c_{\max})}{(1-\gamma)^2}\|Q_k^{c_i} - \hat{Q}_k^{c_i}\|_2\right] + \frac{\sum_{k\in\mathcal{N}_r}}{\sum_{i=0}^m|\mathcal{N}_{c_i}|+|\mathcal{N}_r|}\left[\frac{1}{\alpha}\mathbb{E}_{s\sim S}\big(D_{KL}(\pi_{\theta_p}^*||\pi_{\theta_p^k})\right.
$$

$$
\left. - D_{KL}(\pi_{\theta_p}^*||\pi_{\theta_p^{k+1}})\big) + \frac{2\alpha r_{\max}^2|\mathcal{S}||\mathcal{A}|}{(1-\gamma)^3} + \frac{3(1+\alpha r_{\max})}{(1-\gamma)^2}\|Q_k^r - \hat{Q}_k^r\|_2\right]
$$

$$
= \frac{\sum_{i=0}^m\sum_{k\in\mathcal{N}_{c_i}} + \sum_{k\in\mathcal{N}_r}}{\sum_{i=0}^m|\mathcal{N}_{c_i}|+|\mathcal{N}_r|}\left[\frac{1}{\alpha}\mathbb{E}_{s\sim S}\big(D_{KL}(\pi_{\theta_p}^*||\pi_{\theta_p^k}) - D_{KL}(\pi_{\theta_p}^*||\pi_{\theta_p^{k+1}})\big)\right]
$$

$$
+ \frac{\sum_{i=0}^m\sum_{k\in\mathcal{N}_{c_i}}}{\sum_{i=0}^m|\mathcal{N}_{c_i}|+|\mathcal{N}_r|}\left[\frac{2\alpha c_{\max}^2|\mathcal{S}||\mathcal{A}|}{(1-\gamma)^3}\right] + \frac{\sum_{i=0}^m\sum_{k\in\mathcal{N}_{c_i}}}{\sum_{i=0}^m|\mathcal{N}_{c_i}|+|\mathcal{N}_r|}\left[\frac{3(1+\alpha c_{\max})}{(1-\gamma)^2}\|Q_k^{c_i} - \hat{Q}_k^{c_i}\|_2\right]
$$

$$+ \frac{\sum_{k \in \mathcal{N}_r}}{\sum_{i=0}^{m} |\mathcal{N}_{c_i}| + |\mathcal{N}_r|} \left[ \frac{2\alpha r_{\max}^2 |\mathcal{S}||\mathcal{A}|}{(1-\gamma)^3} + \frac{3(1+\alpha r_{\max})}{(1-\gamma)^2} \|Q_k^r - \hat{Q}_k^r\|_2 \right]$$

$$= \frac{1}{\sum_{i=0}^{m} |\mathcal{N}_{c_i}| + |\mathcal{N}_r|} \left[ \frac{1}{\alpha} \mathbb{E}_{s \sim S} \left( D_{KL}(\pi_{\theta_p}^* || \pi_{\theta_p^0}) - D_{KL}(\pi_{\theta_p}^* || \pi_{\theta_p^{k+1}}) \right) \right]$$

$$+ \frac{\sum_{i=0}^{m} \sum_{k \in \mathcal{N}_{c_i}}}{\sum_{i=0}^{m} |\mathcal{N}_{c_i}| + |\mathcal{N}_r|} \left[ \frac{2\alpha c_{\max}^2 |\mathcal{S}||\mathcal{A}|}{(1-\gamma)^3} + \frac{3(1+\alpha c_{\max})}{(1-\gamma)^2} \|Q_k^{c_i} - \hat{Q}_k^{c_i}\|_2 \right]$$

$$+ \frac{\sum_{k \in \mathcal{N}_r}}{\sum_{i=0}^{m} |\mathcal{N}_{c_i}| + |\mathcal{N}_r|} \left[ \frac{2\alpha r_{\max}^2 |\mathcal{S}||\mathcal{A}|}{(1-\gamma)^3} + \frac{3(1+\alpha r_{\max})}{(1-\gamma)^2} \|Q_k^r - \hat{Q}_k^r\|_2 \right]$$

$$\overset{\{ii\}}{\leq} \frac{1}{\sum_{i=0}^{m} |\mathcal{N}_{c_i}| + |\mathcal{N}_r|} \left[ \frac{1}{\alpha} \mathbb{E}_{s \sim S} \left( D_{KL}(\pi_{\theta_p}^* || \pi_{\theta_p^0}) - D_{KL}(\pi_{\theta_p}^* || \pi_{\theta_p^{k+1}}) \right) \right]$$

$$+ \frac{\sum_{i=0}^{m} \sum_{k \in \mathcal{N}_{c_i}}}{\sum_{i=0}^{m} |\mathcal{N}_{c_i}| + |\mathcal{N}_r|} \left[ \frac{2\alpha Re_{\max}^2 |\mathcal{S}||\mathcal{A}|}{(1-\gamma)^3} + \frac{3(1+\alpha Re_{\max})}{(1-\gamma)^2} \|Q_k^{c_i} - \hat{Q}_k^{c_i}\|_2 \right]$$

$$+ \frac{\sum_{k \in \mathcal{N}_r}}{\sum_{i=0}^{m} |\mathcal{N}_{c_i}| + |\mathcal{N}_r|} \left[ \frac{2\alpha Re_{\max}^2 |\mathcal{S}||\mathcal{A}|}{(1-\gamma)^3} + \frac{3(1+\alpha Re_{\max})}{(1-\gamma)^2} \|Q_k^r - \hat{Q}_k^r\|_2 \right]$$

$$= \frac{1}{\sum_{i=0}^{m} |\mathcal{N}_{c_i}| + |\mathcal{N}_r|} \left[ \frac{1}{\alpha} \mathbb{E}_{s \sim S} \left( D_{KL}(\pi_{\theta_p}^* || \pi_{\theta_p^0}) - D_{KL}(\pi_{\theta_p}^* || \pi_{\theta_p^{k+1}}) \right) \right]$$

$$+ \frac{\sum_{i=0}^{m} \sum_{k \in \mathcal{N}_{c_i}} + \sum_{k \in \mathcal{N}_r}}{\sum_{i=0}^{m} |\mathcal{N}_{c_i}| + |\mathcal{N}_r|} \left[ \frac{2\alpha Re_{\max}^2 |\mathcal{S}||\mathcal{A}|}{(1-\gamma)^3} \right] + \frac{\sum_{k \in \mathcal{N}_r}}{\sum_{i=0}^{m} |\mathcal{N}_{c_i}| + |\mathcal{N}_r|} \left[ \frac{3(1+\alpha Re_{\max})}{(1-\gamma)^2} \|Q_k^r - \hat{Q}_k^r\|_2 \right]$$

$$+ \frac{\sum_{i=0}^{m} \sum_{k \in \mathcal{N}_{c_i}}}{\sum_{i=0}^{m} |\mathcal{N}_{c_i}| + |\mathcal{N}_r|} \left[ \frac{3(1+\alpha Re_{\max})}{(1-\gamma)^2} \|Q_k^{c_i} - \hat{Q}_k^{c_i}\|_2 \right]$$

$$\overset{\{iii\}}{\leq} \frac{1}{\sum_{i=0}^{m} |\mathcal{N}_{c_i}| + |\mathcal{N}_r|} \left[ \frac{1}{\alpha} \mathbb{E}_{s \sim S} \left( D_{KL}(\pi_{\theta_p}^* || \pi_{\theta_p^0}) - D_{KL}(\pi_{\theta_p}^* || \pi_{\theta_p^{k+1}}) \right) \right] + \frac{2\alpha Re_{\max}^2 |\mathcal{S}||\mathcal{A}|}{(1-\gamma)^3}$$

$$+ \frac{\sum_{i=0}^{m} \sum_{k \in \mathcal{N}_{c_i}} + \sum_{k \in \mathcal{N}_r}}{\sum_{i=0}^{m} |\mathcal{N}_{c_i}| + |\mathcal{N}_r|} \left[ \frac{3(1+\alpha Re_{\max})}{(1-\gamma)^2} \frac{\sqrt{(1-\gamma)|\mathcal{S}||\mathcal{A}|}}{\sqrt{K}} \right]$$

$$= \frac{1}{\sum_{i=0}^{m} |\mathcal{N}_{c_i}| + |\mathcal{N}_r|} \left[ \frac{1}{\alpha} \mathbb{E}_{s \sim S} \left( D_{KL}(\pi_{\theta_p}^* || \pi_{\theta_p^0}) - D_{KL}(\pi_{\theta_p}^* || \pi_{\theta_p^{k+1}}) \right) \right]$$

$$+ \frac{3(1+\alpha Re_{\max})}{(1-\gamma)^2} \frac{\sqrt{(1-\gamma)|\mathcal{S}||\mathcal{A}|}}{\sqrt{K}} + \frac{2\alpha Re_{\max}^2 |\mathcal{S}||\mathcal{A}|}{(1-\gamma)^3}$$

$$\overset{\{iv\}}{=} \frac{1}{K\alpha} \mathbb{E}_{s \sim S} \left( D_{KL}(\pi_{\theta_p}^* || \pi_{\theta_p^0}) - D_{KL}(\pi_{\theta_p}^* || \pi_{\theta_p^{k+1}}) \right) + \frac{3(1+\alpha Re_{\max})}{(1-\gamma)^2} \frac{\sqrt{(1-\gamma)|\mathcal{S}||\mathcal{A}|}}{\sqrt{K}}$$

$$+ \frac{2\alpha Re_{\max}^2 |\mathcal{S}||\mathcal{A}|}{(1-\gamma)^3}$$

$$\overset{\{v\}}{\leq} \frac{1}{K\alpha} \mathbb{E}_{s \sim S} \left( D_{KL}(\pi_{\theta_p}^* || \pi_{\theta_p^0}) - D_{KL}(\pi_{\theta_p}^* || \pi_{\theta_p^{k+1}}) \right) + \frac{2\alpha Re_{\max}^2 |\mathcal{S}||\mathcal{A}|}{(1-\gamma)^3} + \frac{3(1+Re_{\max})}{(1-\gamma)^{1.5}} \frac{\sqrt{|\mathcal{S}||\mathcal{A}|}}{\sqrt{K}}$$

$$\overset{\{vi\}}{=} \frac{\sqrt{|\mathcal{S}||\mathcal{A}|}}{(1-\gamma)^{1.5}\sqrt{K}} \mathbb{E}_{s \sim S} \left( D_{KL}(\pi_{\theta_p}^* || \pi_{\theta_p^0}) - D_{KL}(\pi_{\theta_p}^* || \pi_{\theta_p^{k+1}}) \right) + \frac{2Re_{\max}^2 \sqrt{|\mathcal{S}||\mathcal{A}|}}{(1-\gamma)^{1.5}\sqrt{K}}$$

$$+ \frac{3(1+Re_{\max})\sqrt{|\mathcal{S}||\mathcal{A}|}}{(1-\gamma)^{1.5}\sqrt{K}}$$

$$= \frac{\sqrt{|\mathcal{S}||\mathcal{A}|}}{(1-\gamma)^{1.5}\sqrt{K}} \left[ \mathbb{E}_{s \sim S} \left( D_{KL}(\pi_{\theta_p}^* || \pi_{\theta_p^0}) - D_{KL}(\pi_{\theta_p}^* || \pi_{\theta_p^{k+1}}) \right) + 2Re_{\max}^2 + 3 + 3Re_{\max} \right],$$

(16)

where $N_r$ represents the set of policies updated based on maximizing rewards and $|N_r|$ represents the number of steps taken to update based on maximizing rewards. $N_{c_i}$ and $|N_{c_i}|$ represent the set of policy updates based on the $i$-th cost and the number of update steps based on the $i$-th cost, respectively.

$|\mathcal{S}|$ and $|\mathcal{A}|$ are the dimensions of the observation space and the action space, respectively. $K$ is the maximum number of steps for policy updates. Based on the relationship of the policy boundaries updated by natural gradient as shown in Lemma B.1, the relationship of the inequalities $\{i\}$ is obtained. Set the $Re_{\max} = \max\{r_{\max}, c_{\max}\}$, the inequality $\{ii\}$ is derived. Based on the relation $\|Q_k - \hat{Q}_k\|_2 \leq \frac{\sqrt{(1-\gamma)|\mathcal{S}||\mathcal{A}|}}{\sqrt{K}}$, the inequality $\{iii\}$ is obtained. Substitute $K = \sum_{i=0}^{m} |\mathcal{N}_{c_i}| + |\mathcal{N}_r|$, the equation $\{iv\}$ is derived. Where $\alpha = (1-\gamma)^{1.5}/\sqrt{|\mathcal{S}||\mathcal{A}|K} \in (0,1)$, the inequality $\{v\}$ is obtained, and substituting $\alpha = (1-\gamma)^{1.5}/\sqrt{|\mathcal{S}||\mathcal{A}|K}$ results in inequality $\{vi\}$. Based on Eq. (16), it follows that $\mathcal{L}(\pi_{\theta_p}^*) - \mathbb{E}[\mathcal{L}(\pi_{\theta_p^k})] \leq \Theta\left(\frac{\sqrt{|\mathcal{S}||\mathcal{A}|}}{(1-\gamma)^{1.5}\sqrt{K}}\right)$, thus proving Proposition B.2.

**Proposition B.3.** *After updating the policy for $K$ steps according to the method described in Eq.* (8)*, the policy cost converges to a boundary.*

$$\mathbb{E}[\mathcal{L}(\pi_{\theta_p^k})] - \bar{c}_i \leq \Theta\left(\sqrt{\frac{(1-\gamma)|\mathcal{S}||\mathcal{A}|}{K}}\right). \tag{17}$$

***Proof.*** After $K$ updates of the policy, the cost constraint relationship is expressed as follows:

$$\mathbb{E}[\mathcal{L}(\pi_{\theta_p^k})] - \bar{c}_i$$

$$= \frac{1}{\sum_{i=0}^{m} |\mathcal{N}_{c_i}| + |\mathcal{N}_r|}\left[\sum_{i=0}^{m}\sum_{k\in\mathcal{N}_{c_i}} \mathcal{L}_{c_i}(\pi_{\theta_p^k}) + \sum_{k\in\mathcal{N}_r} \mathcal{L}_r(\pi_{\theta_p^k})\right] - \bar{c}_i$$

$$= \frac{1}{\sum_{i=0}^{m} |\mathcal{N}_{c_i}| + |\mathcal{N}_r|}\left[\sum_{i=0}^{m}\sum_{k\in\mathcal{N}_{c_i}} \mathcal{L}_{c_i}(\pi_{\theta_p^k}) + \sum_{k\in\mathcal{N}_r} \mathcal{L}_r(\pi_{\theta_p^k}) - \sum_{i=0}^{m}\sum_{k\in\mathcal{N}_{c_i}} \bar{c}_i - \sum_{k\in\mathcal{N}_r} \bar{c}_i\right]$$

$$= \frac{\sum_{i=0}^{m}\sum_{k\in\mathcal{N}_{c_i}}}{\sum_{i=0}^{m} |\mathcal{N}_{c_i}| + |\mathcal{N}_r|}\left[\mathcal{L}_{c_i}(\pi_{\theta_p^k}) - \bar{c}_i\right] + \frac{\sum_{k\in\mathcal{N}_r}}{\sum_{i=0}^{m} |\mathcal{N}_{c_i}| + |\mathcal{N}_r|}\left[\mathcal{L}_r(\pi_{\theta_p^k}) - \bar{c}_i\right]$$

$$\overset{\{i\}}{\leq} \frac{\sum_{i=0}^{m}\sum_{k\in\mathcal{N}_{c_i}}}{\sum_{i=0}^{m} |\mathcal{N}_{c_i}| + |\mathcal{N}_r|}\left[\mathcal{L}_{c_i}(\pi_{\theta_p^k}) - \bar{c}_i\right]$$

$$= \frac{\sum_{i=0}^{m}\sum_{k\in\mathcal{N}_{c_i}}}{\sum_{i=0}^{m} |\mathcal{N}_{c_i}| + |\mathcal{N}_r|}\left[\left(\mathcal{L}_{c_i}(\pi_{\theta_p^k}) - \bar{\mathcal{L}}_{c_i}(\pi_{\theta_p^k})\right) + \left(\bar{\mathcal{L}}_{c_i}(\pi_{\theta_p^k}) - \bar{c}_i\right)\right]$$

$$= \frac{\sum_{i=0}^{m}\sum_{k\in\mathcal{N}_{c_i}}}{\sum_{i=0}^{m} |\mathcal{N}_{c_i}| + |\mathcal{N}_r|}\left[\mathcal{L}_{c_i}(\pi_{\theta_p^k}) - \bar{\mathcal{L}}_{c_i}(\pi_{\theta_p^k})\right] + \frac{\sum_{i=0}^{m}\sum_{k\in\mathcal{N}_{c_i}}}{\sum_{i=0}^{m} |\mathcal{N}_{c_i}| + |\mathcal{N}_r|}\left[\bar{\mathcal{L}}_{c_i}(\pi_{\theta_p^k}) - \bar{c}_i\right].$$

$$\tag{18}$$

When the policy is updated based on rewards, the policy satisfies the cost restriction fully, resulting in the inequality $\{i\}$. Eq. (18) is further simplified as shown in Eq. (19).

$$\mathbb{E}[\mathcal{L}(\pi_{\theta_p^k})] - \bar{c}_i$$

$$\leq \frac{\sum_{i=0}^{m}\sum_{k\in\mathcal{N}_{c_i}}}{\sum_{i=0}^{m} |\mathcal{N}_{c_i}| + |\mathcal{N}_r|}\left[\mathcal{L}_{c_i}(\pi_{\theta_p^k}) - \bar{\mathcal{L}}_{c_i}(\pi_{\theta_p^k})\right] + \frac{\sum_{i=0}^{m}\sum_{k\in\mathcal{N}_{c_i}}}{\sum_{i=0}^{m} |\mathcal{N}_{c_i}| + |\mathcal{N}_r|}\left[\bar{\mathcal{L}}_{c_i}(\pi_{\theta_p^k}) - \bar{c}_i\right]$$

$$\overset{\{i\}}{\leq} \frac{\sum_{i=0}^{m}\sum_{k\in\mathcal{N}_{c_i}}}{\sum_{i=0}^{m} |\mathcal{N}_{c_i}| + |\mathcal{N}_r|}\left[\mathcal{L}_{c_i}(\pi_{\theta_p^k}) - \bar{\mathcal{L}}_{c_i}(\pi_{\theta_p^k})\right]$$

$$\overset{\{ii\}}{\leq} \frac{\sum_{i=0}^{m}\sum_{k\in\mathcal{N}_{c_i}}}{\sum_{i=0}^{m} |\mathcal{N}_{c_i}| + |\mathcal{N}_r|}\|Q_k^{c_i} - \bar{Q}_k^{c_i}\|_2$$

$$\overset{\{iii\}}{\leq} \frac{\sum_{i=0}^{m}\sum_{k\in\mathcal{N}_{c_i}}}{\sum_{i=0}^{m} |\mathcal{N}_{c_i}| + |\mathcal{N}_r|}\sqrt{\frac{(1-\gamma)|\mathcal{S}||\mathcal{A}|}{K}}$$

$$\leq \frac{1}{2}\sqrt{\frac{(1-\gamma)|\mathcal{S}||\mathcal{A}|}{K}}.$$

$$\tag{19}$$

After $K$ steps of policy updates from lines 14 to 17 in Algorithm 1, assuming that the condition $\bar{\mathcal{L}}_{c_i}(\pi_{\theta_p^k}) - \bar{c}_i \leq 0$, the inequality relation of equation $\{i\}$ is derived. Based on the relation [70] $\mathcal{L}_{c_i}(\pi_{\theta_p^k}) - \bar{\mathcal{L}}_{c_i}(\pi_{\theta_p^k}) \leq \|Q_k^{c_i} - \bar{Q}_k^{c_i}\|_2$, the inequality relation of equation $\{ii\}$ is derived. From the inequality relation $\|Q_k - \hat{Q}_k\|_2 \leq \frac{\sqrt{(1-\gamma)|\mathcal{S}||\mathcal{A}|}}{\sqrt{K}}$, inequality $\{iii\}$ is obtained. Based on Eq. (19), $\mathbb{E}[\mathcal{L}(\pi_{\theta_p^k})] - \bar{c}_i \leq \Theta\left(\sqrt{\frac{(1-\gamma)|\mathcal{S}||\mathcal{A}|}{K}}\right)$, thus proving Proposition B.3.

**Remark B.4.** *The proportion (C/A ratio) between the update iterations allocated to reward maximization and cost minimization influences the final optimal policy $\mathbb{E}[\mathcal{L}(\pi_{\theta_p^k})]$.*

**Proof.** Based on Eq. (16), the relationship between the final policy $\pi_{\theta_p^k}$ after $k$ iterations and the optimal policy $\pi_{\theta_p^*}$ is derived.

$$
\begin{aligned}
&\mathcal{L}(\pi_{\theta_p}^*) - \mathbb{E}[\mathcal{L}(\pi_{\theta_p^k})] \\
&\leq \frac{\sqrt{|\mathcal{S}||\mathcal{A}|}}{(1-\gamma)^{1.5}\sqrt{K}}\left[\mathbb{E}_{s\sim S}\big(D_{KL}(\pi_{\theta_p}^*||\pi_{\theta_p^0}) - D_{KL}(\pi_{\theta_p}^*||\pi_{\theta_p^{k+1}})\big) + 2Re_{\max}^2 + 3 + 3Re_{\max}\right] \\
&= \frac{\sqrt{|\mathcal{S}||\mathcal{A}|}}{(1-\gamma)^{1.5}\sqrt{K}}\left[\left(\sum_{s\sim S}\pi_{\theta_p}^*\log\frac{\pi_{\theta_p}^*}{\pi_{\theta_p^0}} - \sum_{s\sim S}\pi_{\theta_p}^*\log\frac{\pi_{\theta_p}^*}{\pi_{\theta_p^{k+1}}}\right) + 2Re_{\max}^2 + 3 + 3Re_{\max}\right] \\
&= \frac{\sqrt{|\mathcal{S}||\mathcal{A}|}}{(1-\gamma)^{1.5}\sqrt{K}}\left[\sum_{s\sim S}\pi_{\theta_p}^*\left(\log\frac{\pi_{\theta_p}^*}{\pi_{\theta_p^0}} - \log\frac{\pi_{\theta_p}^*}{\pi_{\theta_p^{k+1}}}\right) + 2Re_{\max}^2 + 3 + 3Re_{\max}\right] \\
&= \frac{\sqrt{|\mathcal{S}||\mathcal{A}|}}{(1-\gamma)^{1.5}\sqrt{K}}\left[\sum_{s\sim S}\pi_{\theta_p}^*\log\frac{\pi_{\theta_p^{k+1}}}{\pi_{\theta_p^0}} + 2Re_{\max}^2 + 3 + 3Re_{\max}\right] \\
&= \frac{\sqrt{|\mathcal{S}||\mathcal{A}|}}{(1-\gamma)^{1.5}\sqrt{K}}\left(\sum_{s\sim S}\pi_{\theta_p}^*\log\frac{\pi_{\theta_p^{k+1}}}{\pi_{\theta_p^0}}\right) + \frac{\sqrt{|\mathcal{S}||\mathcal{A}|}}{(1-\gamma)^{1.5}\sqrt{K}}\left(2Re_{\max}^2 + 3 + 3Re_{\max}\right).
\end{aligned}
$$
(20)

In the above Eq. (20), $\mathcal{S},\mathcal{A},\gamma$, and $Re_{\max}$ are constants. Therefore, the deviation between the policy after the $K$ update steps and the optimal policy depends not only on $\frac{\sqrt{|\mathcal{S}||\mathcal{A}|}}{(1-\gamma)^{1.5}\sqrt{K}}$, but also on the policy $\frac{\pi_{\theta_p^{k+1}}}{\pi_{\theta_p^0}}$, which is closely influenced by the number of update steps allocated to the minimization of costs and maximization of rewards. The proportion (C/A ratio) between the update iterations allocated to reward maximization and cost minimization influences the final optimal policy. We define C/A ratio$=\frac{\sum_i^m |\mathcal{N}_{c_i}|}{\sum_i^m |\mathcal{N}_{c_i}| + |\mathcal{N}_r|}$ as the proportion of update steps dedicated to cost minimization relative to the total number of policy, and conduct comprehensive evaluations of its effect through ablation experiments.

## C  Practical Algorithm

To facilitate the understanding of the CHPO algorithm's implementation, we provide its pseudo-code shown in Algorithm 1. After estimating the reward and cost advantage functions $\hat{A}_t^r$ and $\hat{A}_t^{c_i}$, we first employ Eq. (6) and Eq. (7) to update the cost state-value function $V_{\phi_{c_i}}^{c_i}(s)$ and the reward state-value function $V_{\phi_r}^r(s)$, respectively. Subsequently, to more accurately assess unsafe situations, we compute the cost return without the discount factor $\gamma$ as $\bar{\mathcal{L}}_{c_i}(\pi_{\theta_p^k}) = \mathbb{E}_{\tau\sim\pi_{\theta_p^k}}[\sum_{t=0}^{\infty}c_{i,t}]$ in Eq. (5) to decide whether cost constraints are satisfied. When the cost constraints are satisfied, the parameterized policy $\pi_{\theta_p}$ is updated according to Eq. (12) with the objective of maximizing the reward. If the cost constraints are not satisfied, the update objective of the parameterized policy $\pi_{\theta_p}$ is to minimize the cost through Eq. (13). In summary, we evaluate whether the cost constraints are satisfied and update the policy network parameters $\theta_p$, including $\theta_d$ and $\theta_c$, in the policy update step.

---
**Algorithm 1** CHPO
---
1: **Input:** The tuple $(\mathcal{S}, \mathcal{A}_p, C, P, r, \rho_0, \gamma)$
2: **Output:** Policy networks parameters $\theta_p$ including discrete policy parameters $\theta_d$ and continuous policy parameters $\theta_c$;
3: Parameters for the reward and cost state-value $\phi_r, \phi_{c_i}$;
4: **for** *each epoch* **do**
5:     Run the policy $\pi_{\theta_p}$ for $T$ timesteps, collecting the buffer $D = \{(s, a_d, a_c, r, c_i)_t\}_{t=0}^{T}$;
6:     **for** *each batch* **do**
7:         Sample a batch of $(s_t, a_{d,t}, a_{c,t}, r_t, c_{i,t})$ from buffer $D$;
8:         Estimate reward and cost advantages $\hat{A}_t^r$ and $\hat{A}_t^{c_i}$;
9:         *// State-value update step:*
10:        Update the cost state-value $\phi_{c_i}$ via the Eq. (6);
11:        Update the reward state-value $\phi_r$ via the Eq. (7);
12:        *// Policy update step:*
13:        Compute the cost return without the discount factor $\gamma$
            $\bar{\mathcal{L}}_{c_i}(\pi_{\theta_p^k}) = \mathbb{E}_{\tau \sim \pi_{\theta_p^k}}[\sum_{t=0}^{\infty} c_{i,t}]$;
14:        **if** $\bar{\mathcal{L}}_{c_i}(\pi_{\theta_p^k}) \leq \bar{c}_i$ **then**
15:           The policy $\pi_{\theta_p}$ is updated by maximizing the reward as described in Eq. (12),
16:        **else**
17:           The policy $\pi_{\theta_p}$ is updated by minimizing the cost through Eq. (13),
18:        **end if**
19:     **end for**
20: **end for**
---

## D  Experimental Details

### D.1  Task

**Task.** To evaluate the performance of the CHPO algorithm in various safety tasks with hybrid action spaces across different domains, we select three widely adopted tasks including *Moving* [2, 22, 33, 60], *Sliding* [2, 60], and *HardMove* [22, 33, 60] from DI-engine [60] and create a *Parking* task requiring safety considerations as the experimental tasks in this work. To the best of our knowledge, these tasks in the DI-engine are not well-suited for constrained hybrid-action RL tasks. Therefore, to support the further study of constrained hybrid-action RL algorithms, we incorporate danger areas and costs into these tasks of DI-engine. Additionally, to enhance the comprehensiveness of the experiments, we design a custom *Parking* task that involves selecting both discrete and continuous actions while ensuring collision avoidance. The detailed descriptions of tasks are provided below:

**Moving.** In this task, the goal of the agent is to navigate to the target area while avoiding dangerous areas. The agent can choose from discrete actions such as turn, accel, or break, which are combined with two continuous parameters—acceleration value and steering angle—to determine its movement. The movement of the agent is always in the direction of its current direction. An episode ends if the agent reaches the target area, moves out of the field, or exceeds the maximum step 200. The reward is calculated based on the reduction in distance to the target area between consecutive steps. Additionally, the agent receives a bonus reward if it successfully stops within the target area. A cost is incurred each time the agent enters the dangerous areas. The task parameters are as follows: the field is a square with a side length of 2, the target area is a circle with a radius of 0.1, and the danger zones are multiple circles, each with a radius of 0.07.

**Sliding.** In this scenario, the parameterized action spaces available to the agent and the objective of the agent are identical to those of the *Moving* task. The underlying physics in the *Sliding* task differs, as it considers the conservation of inertia, whereas the *Moving* task does not account for inertia conservation. That is to say, the movement of the agent is determined by the vector sum of two polar vectors, influenced by the current action and the previous movement of the agent. In this task, the agent incurs a cost each time it enters a dangerous area. Aside from this, all parameters and rewards are same between the two tasks.

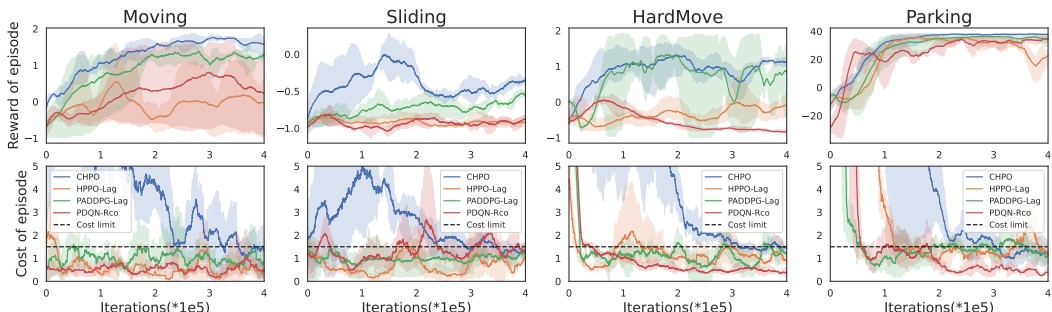

Figure 4: The figure depicts the reward and cost curves of CHPO and the baseline algorithms in the comparison experiment on different cost limits to showcase the performance of each algorithm. The shaded areas on the curves represent the variance obtained from online testing conducted with three random seeds. The cost limit of the four tasks is set at $\bar{c}_i = 1.5$.

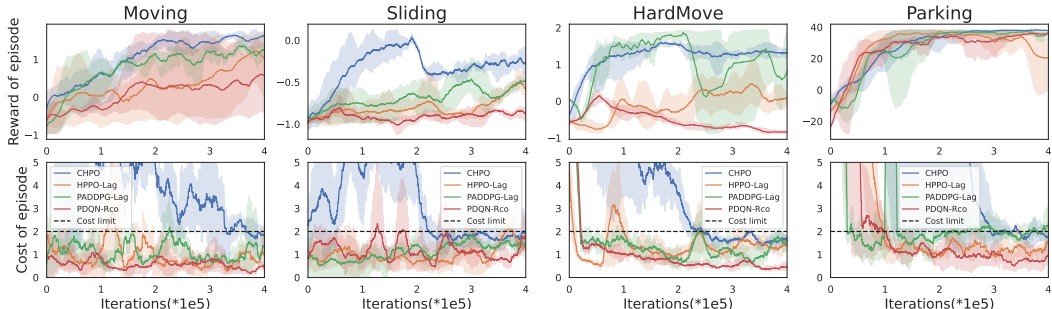

Figure 5: The figure depicts the reward and cost curves of CHPO and the baseline algorithms in the comparison experiment on different cost limits to showcase the performance of each algorithm. The shaded areas on the curves represent the variance obtained from online testing conducted with three random seeds. The cost limit of the four tasks is set at $\bar{c}_i = 2$.

**HardMove.** The task is a discrete-continuous hybrid action space RL task designed to evaluate the performance of RL algorithms in high-dimensional hybrid action spaces. In this task, the agent controls $n$ uniformly distributed actuators with the objective of using their actions to move the agent to the target area while avoiding dangerous areas. Each actuator has two options—on and off—resulting in $2^n$ discrete actions. For actuators that are turned on, the agent must select continuous parameters specifying the distance to move. As the dimensionality of the discrete action space increases exponentially with $n$, and each discrete action is intricately linked to its corresponding continuous parameter, the task presents significant challenges for the scalability and adaptability of constrained hybrid-action RL algorithms. The reward comprises two components: the first is based on the reduction in distance to the target area, and the second is an additional bonus awarded upon reaching the target area. The agent receives a cost for entering the dangerous areas. The episode terminates if one of the three conditions is filled: the agent stops inside the target area, the agent leaves the field, or the step count is higher than the limit 25. In this task, the field is a square with a side length of 2, the target area is a circle with a radius of 0.1, and the dangerous areas are represented by multiple circles with a radius of 0.1.

**Parking.** The task simulates an RS-curve parking scheme, a commonly used parking method in confined spaces. In the RS-curve parking scheme, discrete actions correspond to the selection of specific curve types, while continuous actions represent the distance traveled along the chosen curve. The scheme effectively models the decomposition of actions required for the parking process. The RS-curve parking trajectory is composed of a series of standardized curves including straight segments and arc segments. Based on the current position of the vehicle and the target parking spot, an appropriate curve combination is selected and these curve types form the discrete action spaces. After selecting a curve type, the movement distance of the vehicle along the curve needs to be determined, which constitutes the continuous action spaces. The objective is to park the vehicle in the target area and avoid collisions with barriers. The rewards in the *Parking* task include the reduction in distance to the target area, the reduction in heading deviation, and a reward for successful parking. The costs include the distance between the vehicle and the barriers, along with a penalty incurred for collisions with barriers. An episode terminates when the vehicle either successfully completes the

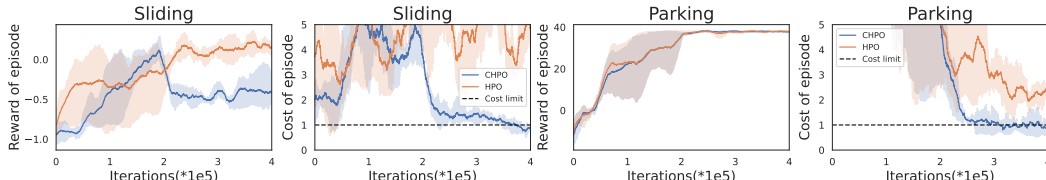

Figure 6: The figure depicts the reward and cost curves in ablation experiments regarding the constraint module. The HPO represents CHPO without the constraint module. The shadowed areas represent the variance of the test results for three random seeds and the cost limit is set at $\bar{c}_i = 1$.

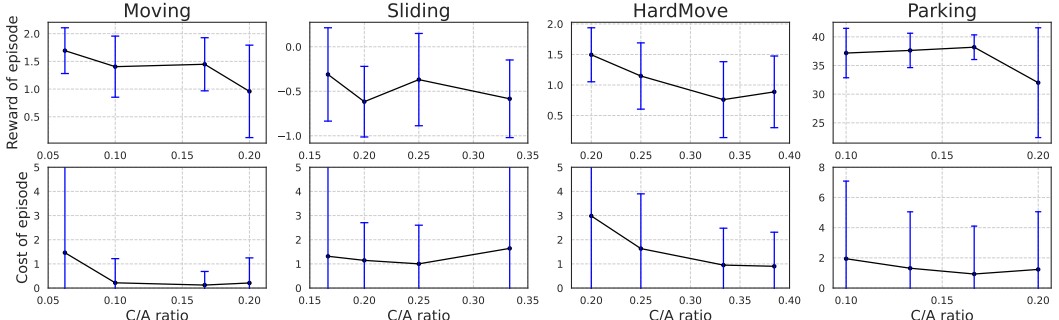

Figure 7: The line chart with variance representations illustrates the performance of CHPO under different ratios of cost minimization policy update counts to all policy update counts (C/A ratio). The results are averaged 120 episodes from 3 random seeds and the cost limit of the four tasks is set at $\bar{c}_i = 1$. Each task is set with four different C/A ratios based on the actual training conditions.

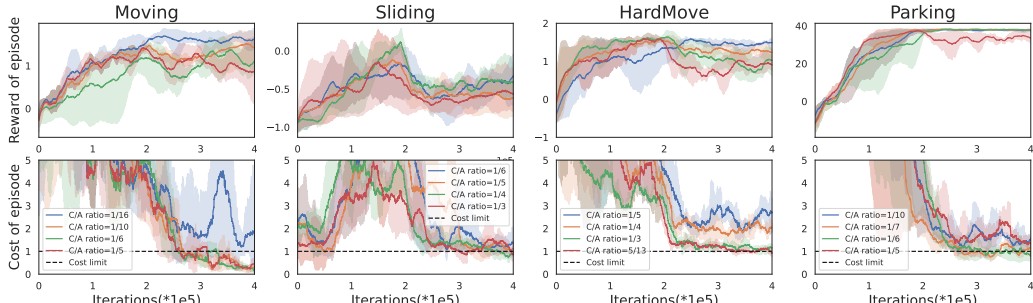

Figure 8: The figure depicts the reward and cost curves of CHPO under different ratios of cost minimization policy update counts to all policy update counts (C/A ratio). Each task is set with four different C/A ratios based on the actual training conditions. The shadowed areas represent the variance of the test results for three different random seeds and the cost limit is set at $\bar{c}_i = 1$.

task, goes out of bounds, or exceeds the maximum step limit of 50. In this task, the dimensions of the vehicle are specified as follows: a width of 2, a length of 4.3, and a front-to-rear axle distance of 2.7. The turning radius of curves is set to 5, while the parking area has a width of 2.5 and a length of 5.3.

## D.2 Experimental Results

To facilitate the analysis of various performance trends during the algorithm training process, we record the reward and cost curves for CHPO and baseline algorithms throughout the training process. These experimental results serve as a supplement to the comparative and ablation experiments in the manuscript, offering a clear understanding of the CHPO's performance during the training process.

**Performance on different cost limits.** Fig. 4 and Fig. 5 illustrate the testing curves of the CHPO algorithm and the baseline algorithms for the four constrained hybrid-action RL tasks under different cost limits. As shown in the figures, the test curves of CHPO during training demonstrate stable behavior, with the cost curves under different cost limits consistently remaining within the safety constraints. Additionally, compared with the baseline algorithms, the cost curves of CHPO are positioned near their respective cost limits, indicating that the CHPO algorithm effectively adapts to the various constraints without overfitting. At the same time, as the cost limit $\bar{c}_i$ increases from

Table 2: The hyper-parameters of the CHPO algorithm model. Where $s$, $a_d$, and $a_c$ denote the dimensions of the state, discrete action, and continuous action respectively. The batch size is set to 320 for the *Moving* and *Sliding* tasks, and 64 for the *HardMove* and *Parking* tasks.

| Sort | Hyper-parameters | Setting |
|---|---|---|
| State-value($r$) | Number of neurons | $s \times 256 \times 128 \times 64 \times 64 \times 64 \times 1$ |
| | Activation function | ReLu |
| | Number of networks | 1 |
| | Learning rate | 3.00e-04 |
| | Optimizer | Adam |
| State-value($c_i$) | Number of neurons | $s \times 256 \times 128 \times 64 \times 64 \times 64 \times 1$ |
| | Activation function | ReLu |
| | Number of networks | 1 |
| | Learning rate | 3.00e-04 |
| | Optimizer | Adam |
| Policy($\pi_p$) | Number of neurons($encoder$) | $s \times 256 \times 128 \times 64 \times 64$ |
| | Number of neurons($\pi_d$) | $encoder \times 64 \times a_d$ |
| | Number of neurons($\pi_c$) | $encoder \times 64 \times a_c$ |
| | Activation function | ReLu |
| | Number of networks | 2 |
| | Learning rate | 3.00e-04 |
| | Optimizer | Adam |
| Others | Batch size | 320 or 64 |
| | Discount factor $\gamma$ | 0.99 |
| | Clip ratio $\epsilon$ | 0.2 |

1 to 2, the reward curves of the CHPO algorithm exhibit a noticeable improvement across all four tasks compared to the baseline algorithms. The above results demonstrate that CHPO can address constrained hybrid-action RL tasks under various cost constraints, delivering competitive rewards while ensuring compliance with safety constraints.

**Performance with and without the constraint module.** Fig. 6 illustrates the reward and cost curves for the CHPO algorithm and the CHPO algorithm without the constraint module in the *Sliding* and *Parking* tasks. From the results in the figure, it can be observed that HPO achieves higher rewards in tasks with parameterized action spaces by disregarding safety constraints, leading to significant violations. In contrast, the CHPO algorithm effectively learns policies that adhere to safety constraints while maintaining a reasonable trade-off in reward. The above analysis of results shows that CHPO successfully handles safety constraints in hybrid action spaces and provides acceptable rewards.

**Performance on different C/A ratios.** Fig. 8 illustrates the testing curves of the CHPO algorithm for the four constrained hybrid-action RL tasks under different ratios of cost minimization policy update counts to all policy update counts (C/A ratio). The results shown in the figure reveal that smaller C/A ratios often fail to satisfy safety constraints, while larger C/A ratios, although effective in meeting safety constraints, tend to induce significant fluctuations in the reward curves. Furthermore, we observe that across all four tasks, the CHPO algorithm can learn policies that satisfy safety constraints and deliver satisfactory rewards under a wide range of C/A ratios. In addition, the mean and standard deviation results presented in Fig. 7 also illustrate that CHPO can achieve satisfactory rewards while ensuring that the average cost satisfies the safety constraint within a broad range of C/A ratios.

### D.3 Experimental Setting

Experiments are run on machines that consist of AMD Ryzen Threadripper 3960X cores and RTX 3090. We provide a detailed explanation of the experimental tasks in Section D.1. Table 2 displays the parameters of the neural network model utilized in our CHPO algorithm. Additionally, detailed configuration information for the testing environment is provided in the code appendix. You can refer to the README file in the appendix code for instructions on installing and configuring the training and testing environment for the CHPO model.

## E   Impact and Limitation Statements

**Impact Statements:** This paper presents work aiming to advance the field of hybrid-action RL, and we believe the study can significantly benefit constrained hybrid-action policy optimization for RL. Moreover, this work can provide a theoretical foundation for certain applications, such as

parameterized safe path planning in autonomous driving and parameterized safe planning and control for robots (agents). To the best of our knowledge, our work cannot be misused in any way to cause any form of negative social impact.

**Limitation Statements:** The study currently focuses on online interactions within simulation experiments, which often suffer from low data sampling efficiency. In the future, we aim to deploy our method in real-world applications and explore offline RL methods for constrained hybrid-action policy optimization.

