# OpenReview forum: "CHPO: Constrained Hybrid-action Policy Optimization for Reinforcement Learning"
_NeurIPS.cc/2025/Conference — NeurIPS 2025 poster_

### Official Review · Reviewer_HetW · 2025-06-14

**Clarity:** 4
**Significance:** 3
**Originality:** 3
**Rating:** 5
**Confidence:** 4

**Summary:**

This paper introduces the Constrained Parameterized-action Markov Decision Process (CPMDP) framework to formalize the problem of maximizing rewards under safety constraints in such hybrid action spaces. They propose the Constrained Hybrid-action Policy Optimization (CHPO) algorithm, which employs a constrained actor-critic architecture with separate critic networks for reward and cost estimation, and dynamically adjusts policy updates. Theoretical analyses demonstrate that CHPO converges to near-optimal solutions while guaranteeing safety constraint satisfaction, with convergence bounds derived for both policy and cost.

**Questions:**

(1)Theoretical proofs assume accurate state-value function approximation. How does CHPO perform when critic networks have high estimation errors (common in high-dimensional or noisy environments)? Are there ablation studies on critic architecture depth or noise injection? Perhaps adding an ablation test with noise cost/reward signals (e.g., adding Gaussian noise to $c_{i,t}$) to verify the robustness can solve this problem. And please consider how the approximation error affects the derived convergence boundary (e.g., the sensitivity of $\|Q_k - \hat{Q}_k\|_2$ in Proposition 4.3).

(2) Acronyms "C/A ratio" is introduced in Remark 4.5 but not defined until Appendix B.4.

(3) The paper states that CHPO is "inspired by HPPO [2]" but does not clearly contrast their architectures. It makes me a little bit confused to identify how CHPO differ from HPPO design.

**Ethical Concerns:**

["NO or VERY MINOR ethics concerns only"]

**Final Justification:**

The experiments and data supplemented by the author have made the conclusion more reliable. The responses have resolved most of my doubts. I have no other questions and decide to increase the score after careful consideration.

**Limitations:**

The dual actor-critic architecture (discrete/continuous actors, reward/cost critics) introduces higher computational complexity compared to single-modality constrained RL methods. The paper does not explicitly analyze computational efficiency (e.g., training time, parameter count), which could be a concern for real-time applications. Baselines like HPPO-Lag or PDQN-Rco may be more lightweight, though less effective.

**Paper Formatting Concerns:**

(1) The paper uses inconsistent formats for referencing equations (e.g., Eq. (5) (with parentheses) and Eq. 5 (without parentheses)). Some spaces in the citation format have been ignored, such as the citation "Fig.6" in Section 5.3. Please check similar issues.

(2) Remark 4.5 in the text uses a formatting style that does not align with the naming conventions of other structured elements (e.g., Lemma 4.2, Proposition 4.3).

**Quality:**

3

**Strengths And Weaknesses:**

Strengths:

(1) The paper is well-written and well-organized. Technical details (e.g., actor-critic architecture, policy update rules) are clearly articulated.

(2) Adequate theories and complete proof processes reflect the theoretical rationality of this method, which is a significant advantage in a field often dominated by empirical methods.

Weaknesses：

 （1）While the baselines (HPPO-Lag, PDQN-Rco) are adapted from existing hybrid-action RL methods, they may not represent the strongest possible constrained RL baselines for hybrid spaces. For instance, integrating modern constrained RL techniques (e.g., CVPO, RCPO) with hybrid action frameworks could provide stiffer competition. The paper’s claim of “first attempt” to address hybrid actions and constraints is valid, but stronger baselines would strengthen the novelty argument.

(2) The convergence proofs rely on idealized assumptions, such as perfect state-value estimation. For example, in Lemma B.1 and Proposition B.2, assume that $V^r(s)$ and $V^{c_i}(s)$ are accurately approximated without estimation errors. The theoretical bounds of the paper (e.g., $\Theta(\frac{\sqrt{|S||A|}}{(1-\gamma)^{1.5}K})$) depend on these idealized conditions, which may limit the practical scalability.

---

> ### Author Rebuttal · Authors · 2025-07-30
>
> We appreciate the thoughtful review and the constructive feedback. We hope the following clarifications and experiments address your questions.
>
> **Q1: Stronger baselines would strengthen the novelty argument.**
>
> Thanks for the kind suggestion. RCPO has been covered in our baseline PDQN-Rco. The Lagrangian dual method we adopt is conceptually similar to CVPO in addressing constrained problems; however, CVPO first computes a variational distribution and then solves the parameterized network based on this distribution, which introduces additional network estimation errors and is not well-suited for constrained hybrid-action policy optimization. We have discussed CVPO in our paper.
>
> **Q2: The theoretical bounds of the paper (e.g., $\Theta \left(\frac{\sqrt{|\mathcal{S}||\mathcal{A}|}}{(1-\gamma)^{1.5}\sqrt{K}}\right)$) depend on these idealized conditions, which may limit the practical scalability.**
>
> Thanks for the kind suggestion. Although our theoretical analysis is based on the assumption of accurate value estimation, we have already incorporated scaling in the analysis, as shown in the inequalities (i), (ii), (iii), and (v) in Eq. (16). Intermittent input noise affects training stability, while persistent noise slows down the convergence rate, but it does not have a significant impact on the final results.
>
> **Q3: How does CHPO perform when critic networks have high estimation errors (common in high-dimensional or noisy environments)?**
>
> Thank you for bringing our attention to the robustness. We add ablation tests with noisy cost and reward signals, where Gaussian noise is injected into the cost and reward signals to evaluate the robustness of CHPO when the critic networks exhibit higher estimation errors. The experimental results of *Parking* are presented below, showing the results under 1% and 5% persistent noise as well as random noise.
> | Gaussian noise |     1%     |     5%     |   random   |      0     |
> |:--------------:|:----------:|:----------:|:----------:|:----------:|
> |     Reward     | 37.27±3.03 | 38.17±2.85 | 37.28±3.26 | 37.63±3.04 |
> |      Cost      |  0.89±3.01 |  0.80±2.79 | 0.78±3.16  | 0.93±3.17  |
>
> Based on the above experiments, we conclude that sporadic random noise does not affect the final results but causes fluctuations during training, while persistent noise slows down the convergence rate without impacting the final stable convergence. Moreover, in theoretical analysis, we apply scaling as shown in the inequalities (i), (ii), (iii), and (v) of Eq. (16). Based on both experimental and theoretical analysis, we believe that the approximation errors do not affect the final stable convergence.
>
> **Q4: Acronyms "C/A ratio" is introduced in Remark 4.5 but not defined until Appendix B.4.**
>
> Thanks for the kind suggestion. We have added the definition of the C/A ratio at line 246 of the manuscript, i.e.,“We define C/A ratio $= \frac{\sum_{i}^{m}|\mathcal{N}{c_i}|}{\sum_{i}^{m}|\mathcal{N}\_{c_i}| + |\mathcal{N}\_r|}$ as the proportion of update steps dedicated to cost minimization relative to the total number of policy updates.”
>
> **Q5: The paper states that CHPO is "inspired by HPPO [2]" but does not clearly contrast their architectures. It makes me a little bit confused to identify how CHPO differ from HPPO design.**
>
> Thanks for the comment. HPPO is a policy optimization method designed for hybrid action spaces, focusing primarily on reward maximization. However, it does not account for safety constraints, which often leads to cost violations when applied to safety-critical tasks.
>
> CHPO builds upon the hybrid-action policy optimization framework of HPPO by introducing Constrained Parameterized-action Markov Decision Process (CPMDP), where safety constraints are explicitly modeled through a cost function $C$. CHPO proposes a constrained hybrid-action actor-critic architecture, consisting of two actors and two critics, where the two critics are responsible for learning the reward and cost state-value functions, respectively. During policy updates, CHPO accounts for safety constraints and rewards, employing a two-stage update mechanism of reward maximization and cost minimization, whereas HPPO optimizes solely for rewards. Although CHPO draws inspiration from HPPO in value function design, its core innovations lie in introducing CPMDP, the constrained hybrid-action actor-critic architecture, and constrained hybrid-action policy optimization, that are absent in HPPO design.
>
> **Q6: The paper does not explicitly analyze computational efficiency (e.g., training time, parameter count), which could be a concern for real-time applications.**
>
> Thank you for bringing our attention to the efficiency. The training time and parameter count of CHPO and the baselines are as follows:
>
> |   Method   | Runtime | Parameter count | Memory | Convergence steps |
> |:----------:|:-------:|:---------------:|:------:|:-----------------:|
> |  HPPO-Lag  |  2.42ms |      167623     |  539MB |      ≥2.5*1e5      |
> | PADDPG-Lag |  1.02ms |      22599      |  481MB |      ≥3.5*1e5      |
> |  PDQN-Rco  |  1.51ms |      67823      |  473MB |      ≥3.0*1e5      |
> |    CHPO    |  2.43ms |      167623     |  503MB |      ≥3.5*1e5      |
>
> Although CHPO has a longer runtime and a larger number of parameters compared to the baselines, their memory usage is similar, and CHPO achieves better performance.
>
> **Q7: The paper uses inconsistent formats and some spaces in the citation format have been ignored.**
>
> Thank you for your careful review. We have checked similar issues and revised the formatting of equation and figure references in the manuscript: “Eq. 5” has been changed to “Eq. (5)” at line 613, and “Fig.6” has been changed to “Fig. 6” at line 334. Additionally, we have added a period at the end of Eq. (20) between lines 599 and 600.
>
> **Q8: Remark 4.5 in the text uses a formatting style that does not align with the naming conventions of other structured elements (e.g., Lemma 4.2, Proposition 4.3).**
>
> Thank you for your careful review. We have adjusted the formatting style of Remark 4.5 in the text to match the naming conventions of Lemma 4.2 and Proposition 4.3.

---

> > ### Comment · Reviewer_HetW · 2025-08-07
> >
> > Thank you for the author's detailed response. The experiments and data supplemented by the author have made the conclusion more reliable. This part of the response and other responses have resolved most of my doubts. I have no other questions and will decide whether to increase the score after careful consideration.

---

> > > ### Author Response · Authors · 2025-08-08
> > >
> > > Thank you for taking the time to re-evaluate our work and for your thoughtful follow-up. We truly appreciate your acknowledgment of our clarifications.

---

### Official Review · Reviewer_jBVV · 2025-06-30

**Clarity:** 3
**Significance:** 2
**Originality:** 2
**Rating:** 4
**Confidence:** 4

**Summary:**

The paper introduces Constrained Hybrid-action Policy Optimization (CHPO), a novel algorithm for reinforcement learning in parameterized action spaces under safety (cost) constraints. The authors first formulate the problem as a Constrained Parameterized‐action Markov Decision Process (CPMDP), extending the standard PAMDP framework to include cost functions alongside rewards. They then propose a two‐critic, two‐actor hybrid actor–critic architecture: one critic and actor pair for rewards, and another for costs. By dynamically switching between reward maximization and cost minimization depending on whether the current policy violates safety thresholds, CHPO directly optimizes the constrained objective without relying on auxiliary Lagrange multipliers. Theoretical analysis proves that CHPO converges to an optimal safe policy within a bounded error that scales with the state/action dimensionality and number of updates. Empirically, CHPO consistently maintains costs below the specified limits while achieving higher or comparable rewards relative to four baselines.

**Questions:**

1. What is the run-time/memory overhead of maintaining two critics and two actors compared to standard PAMDP methods?
2. Evaluations are limited to four relatively low-dimensional tasks; performance on more complex robotics or vision-based environments remains untested.
3. Convergence bounds assume accurate value estimates and tabular |S|,|A|. However, since |A| includes continuous parameters, how is this handled in your analysis?
4. In theoretical analysis, the author use the notation of |A| and $\pi_{\theta_p}$, which correspond to a standard MDP formulation. How is the PAMDP structure incorporated into the analysis?
5. What is the backbone of CHPO? Is it based on HPPO, PDQN, or PADDPG?
6. In the empirical section, the authors compare CHPO with HPPO-Lag, PADDPG-Lag, and PDQN-Rco. Why were HPPO-Rco, PADDPG-Rco, and PDQN-Lag not included for a more balanced comparison?

**Ethical Concerns:**

["NO or VERY MINOR ethics concerns only"]

**Final Justification:**

I would keep my original score of “borderline accept,” as my major concerns remain: their theory does not leverage the special architecture of PAMDPs, and they ultimately did not present experimental results on the current SOTA PAMDP algorithm, DLPA. That said, this is the first paper to consider safety RL in the PAMDP domain, which does provide a degree of novelty.

**Limitations:**

Yes, in Appendix E.

**Paper Formatting Concerns:**

The paper would benefit from visual illustrations of each benchmark.

**Quality:**

3

**Strengths And Weaknesses:**

Strengths
- First to define the CPMDP framework for constrained hybrid‐action RL, neatly integrating cost constraints into PAMDPs.
- Rigorous theoretical bounds lend confidence in the stability and safety of learning.


Weaknesses
- The method relies heavily on the PAMDP structure, but the theoretical analysis does not appear to explicitly address or leverage this structure.
- The evaluated environments lack diversity and are quite similar in nature, which limits the generalizability of the results.

---

> ### Author Rebuttal · Authors · 2025-07-30
>
> Thank you for your valuable feedback. We appreciate the time and effort you have dedicated to reviewing our paper. We will address the weaknesses and questions you raised below.
>
> **Q1: The theoretical analysis does not appear to explicitly address or leverage PAMDP structure.**
>
> Thanks for the comment. Based on PAMDP, we divide the policy network into a discrete policy $\pi_{\theta_d}$ and a continuous policy $\pi_{\theta_c}$, which share the same backbone but differ in the output heads: one outputs discrete actions, while the other outputs continuous actions. For clarity, we denote their parameters as $\theta_d$ and $\theta_c$, respectively. Since $\theta_d$ and $\theta_c$ share the same optimization objective, we unify them as $\theta_p$. Our theoretical analysis is consistently conducted on the hybrid policy parameter $\theta_p$ derived from PAMDP. We define $|\mathcal{A}| := |\mathcal{A}_d| + |\mathcal{A}_c|$, which unifies the discrete and continuous action spaces in PAMDP into $|\mathcal{A}|$ for theoretical analysis. We have further clarified in the paper the relationship among $\theta_p$, $|\mathcal{A}|$, and PAMDP in theoretical analysis.
>
> **Q2: The evaluated environments lack diversity and are quite similar in nature.**
>
> Thanks for the kind suggestion. Currently, there is no existing simulation environment specifically designed for the constrained hybrid-action setting, so we have to modify and construct tasks for evaluation. We are applying our approach to real-world exploration environments, specifically to the RM65 robotic arm, where high-level motion primitives (fixed discrete actions) are manually designed, such as Lift (raising the end-effector upward) and Twist (rotating the gripper or object along a certain axis). The execution details of each discrete action are controlled by continuous parameters. We are in the process of applying CHPO to this robotic arm to accomplish hybrid-action tasks while ensuring safety.
>
> **Q3: What is the run-time/memory overhead of maintaining two critics and two actors compared to standard PAMDP methods?**
>
> Thank you for bringing our attention to the efficiency. We have recorded the run-time and memory usage of the standard PAMDP method (HPPO) and CHPO, as shown in the table below. Additionally, we have supplemented the run-time and memory usage of other baseline algorithms.
> |   Method   | Runtime | Memory | Parameter count |
> |:----------:|:-------:|:------:|:---------------:|
> |    HPPO   |  1.66ms |  490MB |      113222     |
> |  HPPO-Lag  |  2.42ms |  539MB |      167623     |
> | PADDPG-Lag |  1.02ms |  481MB |      22599      |
> |  PDQN-Rco  |  1.51ms |  473MB |      67823      |
> |    CHPO    |  2.43ms |  503MB |      167623     |
>
> **Q4: Evaluations are limited to four relatively low-dimensional tasks; performance on more complex robotics or vision-based environments remains untested.**
>
> Thank you for pointing this out. Our experiments are currently limited to low-dimensional tasks because there is no existing simulation environment for the hybrid-action settings. We modify and build tasks for evaluation. We are actively applying our approach to real-world RM65 robotic arm exploration. In this setting, the discrete actions of the robotic arm include operations such as lift, twist, and grasp, while the continuous actions correspond to the execution parameters for each discrete action. We perform constrained hybrid-action policy optimization on the robotic arm to accomplish tasks while ensuring safety.
>
> **Q5: Convergence bounds assume accurate value estimates and tabular |S|,|A|. However, since |A| includes continuous parameters, how is this handled in your analysis?**
>
> According to related work [1], the boundary of the critic network is related to the dimensions of $|\mathcal{S}|$ and $|\mathcal{A}|$, and the relationship holds in both discrete and continuous spaces $||Q_k - \hat{Q}_k ||_2  \leq \frac{\sqrt{(1-\gamma)|\mathcal{S}||\mathcal{A}|}}{\sqrt{K}}$. Building on [1][2], this work further defines $|\mathcal{A}| := |\mathcal{A}_d| + |\mathcal{A}_c|$, which remains applicable to the convergence analysis of our method. We have added clarifications regarding this in the revised manuscript.
>
> Moreover, our work is based on tabular-form derivations, primarily focusing on analyzing and illustrating the decisive parameters affecting the convergence boundary of the CHPO algorithm. Therefore, we have not derived the convergence for network-based representations. If such a derivation is required, one directly replaces the parameterized representation with the weights and biases of the policy network, in which case the convergence boundary will incorporate the estimation error of the policy network. We have added this clarification to the revised manuscript.
>
> **Q6: How is the PAMDP structure incorporated into the analysis?**
>
> Thanks for the comment. Based on the PAMDP framework, the hybrid action space consists of a discrete action space $\mathcal{A}\_d$ and a continuous action space $\mathcal{A}\_c$. We define $|\mathcal{A}|:= |\mathcal{A}\_d| + |\mathcal{A}\_c|$ to represent this hybrid action space for subsequent theoretical analysis. Under the PAMDP framework, we employ a discrete policy $\pi_{\theta_d}$ to output discrete actions and a continuous policy $\pi_{\theta_c}$ to output continuous actions. In the implementation of the CHPO algorithm, these two policies share the same backbone network but are distinguished at the output heads to reduce computational overhead. Since the optimization objectives of $\pi_{\theta_d}$ and $\pi_{\theta_c}$ are identical, we unify them into the hybrid-action policy $\pi_{\theta_p}$ for the subsequent theoretical analysis, where $\theta_p$ is composed of $\theta_d$ and $\theta_c$.
>
> **Q7: What is the backbone of CHPO? Is it based on HPPO, PDQN, or PADDPG?**
>
> Thank you for catching this. CHPO is built upon HPPO. We chose HPPO because, among the three algorithms, HPPO demonstrates superior performance in handling purely hybrid-action tasks. PADDPG addresses hybrid-action tasks by converting the discrete action space into a continuous one, while PDQN separately applies DQN for discrete actions and DDPG for continuous actions. Compared to HPPO, both PDQN and PADDPG involve more complex handling of hybrid action spaces and show inferior performance. We also evaluate the performance of these three algorithms in terms of reward maximization without cost consideration. The results averaged over 3 seeds in *Moving* are as follows:
> | Method |    HPPO   |   PADDPG  |    PDQN   |
> |:------:|:---------:|:---------:|:---------:|
> | Reward | 1.86±0.28 | 1.47±0.54 | 1.64±0.31 |
>
> These results indicate that HPPO performs better on hybrid action tasks when safety constraints are not considered.
>
> **Q8: Why were HPPO-Rco, PADDPG-Rco, and PDQN-Lag not included for a more balanced comparison?**
>
> Thank you for catching this. PDQN cannot be combined with the Lag method to form PDQN-Lag because PDQN consists of two separate components: DQN for selecting discrete actions and DDPG for selecting continuous actions, making it infeasible to integrate the Lag method into both parts simultaneously. Therefore, we choose to combine PDQN with RCPO to form PDQN-Rco. In addition, we combine both HPPO and PADDPG with RCPO and conduct tests in the *Parking* environment. The comparative results under the cost limit $\bar{c}_i = 1$ are as follows:
> | Method | HPPO-Rco    | PADDPG-Rco  | HPPO-Lag    | PADDPG-Lag  | PDQN-Rco   | CHPO                |
> |--------|-------------|-------------|-------------|-------------|------------|---------------------|
> | Reward | 20.04±22.04 | 21.74±16.02 | 20.01±25.21 | 26.66±16.39 | 32.72±7.44 | **37.63±3.04**      |
> | Cost   | 1.55±4.05   | 0.46±2.35   | 0.82±1.26   | 0.94±2.15   | 0.21±0.35  | **0.93±3.17**
>
> Their performance is inferior to PDQN-Rco; therefore, we ultimately select PDQN-Rco as the baseline instead of HPPO-Rco and PADDPG-Rco.
>
> **Reference**
>
> [1] Dalal, G., Szorenyi, B., & Thoppe, G. (2020, April). A tale of two-timescale reinforcement learning with the tightest finite-time bound. In Proceedings of the AAAI Conference on Artificial Intelligence (Vol. 34, No. 04, pp. 3701-3708).
>
> [2] Xu, T., Liang, Y., & Lan, G. (2021, July). Crpo: A new approach for safe reinforcement learning with convergence guarantee. In International Conference on Machine Learning (pp. 11480-11491). PMLR.

---

> > ### Comment · Reviewer_jBVV · 2025-08-05
> >
> > The authors provide clear clarifications on several points: the run-time and memory overhead is small relative to HPPO despite maintaining two actors and two critics, the choice of baselines is justified with additional comparisons, and the theoretical assumptions about value estimation bounds are now explicitly connected to existing CRPO-style results. The authors also acknowledge the limited diversity of environments and note ongoing work applying the method to real-world robotic tasks, which is understandable given the lack of standardized challenging hybrid-action constrained domains; though I understand this limitation, exploring more diverse or complex domains would still strengthen the paper.
> >
> > However, it remains unclear how the PAMDP structure is explicitly leveraged in the theoretical analysis. In fact, the convergence bounds appear to treat the hybrid action space as a monolithic set |A|, and the same results would hold if PAMDP were replaced by a regular MDP, meaning the parameterized structure is not theoretically exploited.
> >
> > Additionally, I wonder if this approach could be extended to model-based PAMDP methods, like DLPA [1].
> >
> > Overall, these responses address most of my original concerns, though some theoretical details remain unclear. I will keep my original positive score.
> >
> > [1] Zhang, Renhao, et al. "Model-based reinforcement learning for parameterized action spaces." arXiv preprint arXiv:2404.03037 (2024).

---

> > > ### Author Response · Authors · 2025-08-05
> > >
> > > We sincerely thank you for your positive response and thoughtful review. Your suggestions are critical to improving the quality of this work.
> > >
> > > Regarding the derivation of policy convergence bounds under parameterized structures, our work builds upon prior research and leverages the known convergence properties of critic networks in both continuous and discrete action spaces. The derivation shows that the convergence bound is primarily related to the dimensions of the continuous and discrete action spaces, rather than the structure of the policy network itself. However, in constrained hybrid-action settings, the convergence bound is influenced by the number of iterations $K$, the discount factor $\gamma$, and the dimensions of the observation and hybrid action spaces. The increased dimensionality of the hybrid action space results in a looser convergence bound for constrained policy optimization.
> > >
> > > The model-based PAMDP method DLPA learns a parameterized-action-conditioned dynamics model and applies an improved model predictive path integral control to maximize rewards. Our work provides a general-purpose constrained policy optimization method for hybrid action spaces. By applying a two-stage policy optimization objective to the model learning process, DLPA enables safe action prediction. Therefore, our algorithm is also applied within the framework of model-based PAMDP. We have expanded the discussion on constrained policy optimization for model-based PAMDP in the related work section of the manuscript.

---

> > > > ### Comment · Reviewer_jBVV · 2025-08-06
> > > >
> > > > Thanks for the rebuttal. Regarding theoretical analysis, I still remain unconvinced that the PAMDP structure is truly leveraged. Regarding model-based PAMDP (DLPA), I understand that discussion time is limited. However, since DLPA is the current SOTA for PAMDP benchmarks, showing results on even one or two domains by integrating CHPO into the DLPA framework would be highly valuable. If the authors could provide such results, I would be willing to raise my score to “accept.”

---

> ### Author Response · Authors · 2025-08-06
>
> Thank you for your further feedback. We’re grateful for the opportunity to improve our work. We are currently integrating CHPO into the DLPA framework and will provide the corresponding experimental results as soon as possible. If time does not permit, we will include the results in the appendix of the revised manuscript.

---

> ### Author Response · Authors · 2025-08-06
>
> Thank you for your response. Regarding the theoretical analysis, the convergence bounds of RL policies for both continuous and discrete actions are positively correlated with the dimensionality of the observation and action spaces. Although our work is based on the hybrid action PAMDP framework, both value estimation and policy updates are conducted following the principles of proximal policy gradients. Therefore, the theoretical derivation for the hybrid action setting in our work still fundamentally relies on theoretical foundations developed for single action space RL. As a result, apart from the dimensional definition of the hybrid action space, the conclusions we derive remain consistent with the theoretical foundations established in prior work [1].
>
> **Reference**
>
> [1] Dalal, G., Szorenyi, B., & Thoppe, G. (2020, April). A tale of two-timescale reinforcement learning with the tightest finite-time bound. In Proceedings of the AAAI Conference on Artificial Intelligence (Vol. 34, No. 04, pp. 3701-3708).

---

### Official Review · Reviewer_33v4 · 2025-06-30

**Clarity:** 2
**Significance:** 3
**Originality:** 2
**Rating:** 4
**Confidence:** 4

**Summary:**

The paper studies constrained hybrid-action reinforcement learning, where the goal is to learn a safe policy within a parameterized hybrid action space. The authors propose a novel Constrained Hybrid-action Policy Optimization algorithm (CHPO), which optimizes policies directly using a primal approach without Lagrangian multipliers. Experiments on multiple tasks show that CHPO achieves higher rewards while strictly satisfying safety constraints and outperforming existing baselines.

**Questions:**

What the connection is between the proposed algorithm and the hybrid-action setting in this paper?

**Ethical Concerns:**

["NO or VERY MINOR ethics concerns only"]

**Final Justification:**

Most of my concerns are addressed.

**Limitations:**

yes

**Paper Formatting Concerns:**

No formatting concerns.

**Quality:**

3

**Strengths And Weaknesses:**

**Strengths**

1. The paper is well written. The algorithm is simple but intuitively correct and effective.
2. The paper provides a convergence analysis, which helps understand the stability and reliability of the proposed CHPO algorithm.
3. The authors provide enough experiment results to illustrate the advantages of their algorithm compared to existing works.


**Weaknesses**

1. The algorithm requires two actor/critic networks, which doubles the computational burden and makes it difficult to apply to large neural networks, i.e., LLMs.
2. This paper does not demonstrate any algorithmic justification for using a “hybrid-action” setup. Essentially, $\theta_d$ and $\theta_c$ can be combined and treated as a single continuous policy. The authors need to explain what is special about their hybrid setting.
3. The authors should compare to existing contained PPO algorithm [1] and discuss the difference to it.


[1] Jayant, A. K., & Bhatnagar, S. (2022). Model-based safe deep reinforcement learning via a constrained proximal policy optimization algorithm. Advances in Neural Information Processing Systems, 35, 24432-24445.

---

> ### Author Rebuttal · Authors · 2025-07-29
>
> We sincerely thank the reviewer for their insightful feedback, and we’re grateful for the opportunity to improve and clarify our work through answering your detailed questions and concerns.
>
> **Q1: The algorithm requires two actor/critic networks, which doubles the computational burden and makes it difficult to apply to large neural networks, i.e., LLMs.**
>
> Thank you for catching this. The addition of the cost value estimation network indeed increases some computational overhead. However, in our study, the policy network is not composed of two independent networks. Instead, the discrete and continuous action policies share a common backbone network, with two separate output heads generating the discrete actions and the continuous parameters, respectively.
>
> Our work primarily focuses on the constrained policy optimization problem for agents that take structured parameters as input and output actions in hybrid action spaces. If this approach is extended to large neural networks, the constrained policy optimization of our algorithm is still applicable. By leveraging methods such as GRPO [1] and replacing value estimation networks with statistical sampling, the critic networks are removed.
>
> **Q2: This paper does not demonstrate any algorithmic justification for using a “hybrid-action” setup.**
>
> Thanks for the comment. Hybrid-action RL algorithms are designed to handle special scenarios where discrete actions and their associated continuous parameters are selected simultaneously. Conventional RL is limited to either purely discrete or continuous action spaces and cannot address hybrid-action problems. In real-world scenarios, such as a soccer game, the agent selects a discrete action (e.g., dash, turn, tackle, or kick) and determines continuous actions such as power and direction to successfully perform a shooting task. Conventional RL methods do not simultaneously handle the selection of discrete actions and the determination of continuous actions. In contrast, hybrid-action RL policy outputs both the selected discrete action and its corresponding continuous parameters simultaneously.
>
> The parameters $\theta_d$ and $\theta_c$ are not treated as a single continuous policy. Unlike PADDPG, which converts discrete actions into continuous action spaces, our policy network shares a single backbone network with two distinct output heads: one outputs the discrete actions and the other outputs the continuous actions. The differences between the two actor networks arise solely from their separate output heads, and we denote their parameters as $\theta_d$ and $\theta_c$ to distinguish them. Since the optimization objectives of $\theta_d$ and $\theta_c$ are aligned, we unify them under the parameter $\theta_p$ to jointly update both discrete and continuous policies.
>
> In our constrained hybrid-action setting, the agent not only selects both the discrete action and the continuous action simultaneously but also satisfies safety constraints. For example, in the *Parking* task, the agent needs to determine the curve type and its arc length while avoiding collisions with surrounding obstacles.
>
> **Q3: The authors should compare to existing contained PPO algorithm [2] and discuss the difference to it.**
>
> Thanks for the kind suggestion. Compared with the existing contained PPO algorithm, our approach handles constrained policy optimization in hybrid action spaces. The existing contained PPO algorithm focuses on maximizing rewards under safety constraints within a single action space and does not account for hybrid-action scenarios. In contrast, our method achieves constrained policy optimization in discrete–continuous hybrid action spaces. To clearly highlight the differences from [2], we have revised the related work section of the manuscript to explicitly discuss that the “existing constrained PPO algorithm [2] focuses on constrained policy optimization in a single action space and does not consider hybrid-action scenarios.”
>
> **Q4: What the connection is between the proposed algorithm and the hybrid-action setting in this paper?**
>
> Building on the foundation of constrained hybrid-action spaces, our work further addresses the critical issue of safety. CHPO integrates cost constraints into the hybrid-action policy optimization process, ensuring that the policy not only achieves high rewards but also satisfies safety constraints when jointly optimizing discrete actions and continuous actions. In our hybrid-action setting, the agent simultaneously considers both the selection of discrete actions and the determination of continuous actions while ensuring safety. As illustrated in Fig. 1 of the manuscript, conventional hybrid-action RL methods cannot learn safe policies in discrete-continuous hybrid action spaces. Our proposed algorithm specifically targets the problem of constrained policy optimization in the hybrid-action setting.
>
> **Reference**
>
> [1] Shao, Z., Wang, P., Zhu, Q., Xu, R., Song, J., Bi, X., ... & Guo, D. (2024). Deepseekmath: Pushing the limits of mathematical reasoning in open language models. arXiv preprint arXiv:2402.03300.
>
> [2] Jayant, A. K., & Bhatnagar, S. (2022). Model-based safe deep reinforcement learning via a constrained proximal policy optimization algorithm. Advances in Neural Information Processing Systems, 35, 24432-24445.

---

> ### Comment · Reviewer_33v4 · 2025-08-06
>
> Thanks for the responses! Some of my concerns have been addressed, but I still have a question.
>
> > Conventional RL is limited to either purely discrete or continuous action spaces and cannot address hybrid-action problems.
>
> Why? discrete space is a specific case of continuous space. In my understanding, these two spaces can be directly simplified into a single continuous space. I can understand the hybrid setting considered in this paper; I just don't understand why we can't first simply the two spaces into a single continuous one and then directly apply existing constrained PPO algorithm [1].

---

> > ### Author Response · Authors · 2025-08-06
> >
> > Thank you for your detailed and thoughtful feedback. We hope the clarifications below address your concerns.
> >
> > Thank you for catching this. You suggested an alternative approach that predates hybrid-action RL. In early exploration, prior work [1] proposed transforming the discrete action space into a continuous one, effectively simplifying the discrete–continuous hybrid action space into a single continuous space. However, by relaxing the hybrid action space into a continuous set, this approach significantly increases the action space complexity [2], which negatively impacts the reward performance of RL algorithms. Additionally, we conducted a *Parking* experiment using the conventional RL method by simplifying the hybrid action space into a single continuous action space without considering safety constraints, and compared the results against those obtained using the hybrid-action RL algorithm. The results are as follows:
> > | Method | Conventional RL | Hybrid-action RL |
> > |:------:|:---------------:|:----------------:|
> > | Reward |    1.47±0.54    |     1.86±0.28    |
> >
> > From the above experimental results, it is evident that directly simplifying the discrete and continuous spaces into a single continuous space without considering constrained policy optimization leads to worse reward performance compared to the hybrid-action RL algorithm. Therefore, in order to achieve strong reward performance while satisfying safety constraints, we did not apply existing constrained PPO algorithm [3] directly to the relaxed continuous space. Instead, we designed a constrained hybrid-action actor-critic architecture to separately handle discrete and continuous actions, and proposed the Constrained Hybrid-action Policy Optimization (CHPO) algorithm to maximize reward while enforcing safety constraints. To clearly highlight the difference from [3], we revised the related work section of the manuscript to explicitly state "Directly relaxing the hybrid action space into a continuous space significantly increases action space complexity, resulting in poor reward performance, and thus existing constrained PPO algorithm [3] cannot be directly applied."
> >
> > **Reference**
> >
> > [1] Hausknecht, M., & Stone, P. (2015). Deep reinforcement learning in parameterized action space. arXiv preprint arXiv:1511.04143.
> >
> > [2] Xiong, J., Wang, Q., Yang, Z., Sun, P., Han, L., Zheng, Y., ... & Liu, H. (2018). Parametrized deep q-networks learning: Reinforcement learning with discrete-continuous hybrid action space. arXiv preprint arXiv:1810.06394.
> >
> > [3] Jayant, A. K., & Bhatnagar, S. (2022). Model-based safe deep reinforcement learning via a constrained proximal policy optimization algorithm. Advances in Neural Information Processing Systems, 35, 24432-24445.

---

> ### Comment · Reviewer_33v4 · 2025-08-06
>
> > Additionally, we conducted a Parking experiment using the conventional RL method by simplifying the hybrid action space into a single continuous action space without considering safety constraints, and compared the results against those obtained using the hybrid-action RL algorithm.
>
> A bit weird here. I think it would only be fair to compare RL methods that simplify the hybrid action space into a single continuous action space then optimize **with** safety constraints. Why is the comparison done without constraints here?

---

> > ### Author Response · Authors · 2025-08-06
> >
> > Thank you for your further response. We hope the following explanation addresses your concerns.
> >
> > Thank you for pointing this out. The unconstrained comparison was intended to provide an intuitive illustration that simplifying hybrid action space into a single continuous action space increases action space complexity and leads to poorer reward performance. Additionally, in the experimental section of our manuscript, we include the conventional RL method PADDPG-Lag that simplifies the hybrid action space into a single continuous space and then applies constrained policy optimization. The results under the cost limit in the *Moving* environment are as follows:
> > | Method | PADDPG-Lag |    CHPO   |
> > |:------:|:----------:|:---------:|
> > | Reward |  1.30±0.67 | 1.41±0.55 |
> > |  Cost  |  0.36±1.59 | 0.22±0.99 |
> >
> > The experimental results show that directly simplifying the hybrid action space into a single continuous space when incorporating constrained policy optimization leads to inferior performance in both reward and safety compared to CHPO. Results from the other three environments presented in the manuscript further demonstrate that CHPO outperforms the conventional RL method that reduce the hybrid action space to a single continuous space when safety is taken into account.
> >
> > We would be happy to provide any further clarification that may assist in your final evaluation.

---

> > > ### Comment · Reviewer_33v4 · 2025-08-08
> > >
> > > Given that the main difference between this paper's algorithm and [1] lies in the hybrid action space setting, the authors should compare against the algorithm in [1] to demonstrate the negative effect of relaxing the space, rather than comparing to unrelated prior methods.
> > >
> > > By the way, apart from the hybrid setting, are there any other differences between the algorithm in this paper and the one in [1]? Intuitively, the theoretical part seems essentially the same. Considering that the hybrid setting doesn’t seem to introduce additional challenges for the theoretical analysis, am I misunderstanding something?
> > >
> > > [1] Jayant, A. K., & Bhatnagar, S. (2022). Model-based safe deep reinforcement learning via a constrained proximal policy optimization algorithm. Advances in Neural Information Processing Systems, 35, 24432-24445.

---

> > ### Author Response · Authors · 2025-08-07
> >
> > Thank you again for your time and consideration. We wanted to check if there are any further clarifications or details we can provide to assist in your assessment.

---

> > ### Author Response · Authors · 2025-08-08
> >
> > Dear Reviewer,
> >
> > We hope this message finds you well. As the discussion period is nearing its end with **less than one day remaining**, we wanted to ensure we have addressed all your concerns satisfactorily. If there are any additional points or feedback you'd like us to consider, please let us know. Your insights are invaluable to us, and we are eager to address any remaining issues to improve our work.
> >
> > Thank you for your time and effort in reviewing our paper.
> >
> > Sincerely,
> >
> > The Authors

---

> ### Author Response · Authors · 2025-08-09
>
> Thank you for your further response. We hope the following explanation addresses your concerns.
>
> **Q1: Given that the main difference between this paper's algorithm and [1] lies in the hybrid action space setting, the authors should compare against the algorithm in [1] to demonstrate the negative effect of relaxing the space, rather than comparing to unrelated prior methods.**
>
> Thanks for the comment. The work [1] implements constrained policy optimization using Lagrangian multipliers. To apply it to the hybrid action space setting, the discrete action space within the hybrid action space needs to be converted into a continuous action space. This is the same processing idea used by the earlier method PADDPG [2], and PADDPG has already been shown to remain effective under such a transformation. In our study, we also employ Lagrangian multipliers to perform constrained policy optimization after relaxing the hybrid action space into a continuous action space. In essence, PADDPG-Lag is equivalent to work [1], and PADDPG has already been demonstrated to adapt well to hybrid action spaces. Moreover, the network architecture and negative log-likelihood loss used in the model-learning component of the work [1] are not specifically designed for this “relaxation–projection” scenario.
>
> The core of the CHPO algorithm lies in: (i) addressing policy outputs in a hybrid action space to handle discrete–continuous action selection, and (ii) solving the constrained optimization problem using primal policy optimization without introducing additional hyperparameters. The work [1] proposes a model-based safe RL algorithm that leverages a dual method to address safety constraints and achieves good performance. However, there are significant differences between their approach and ours, including: (1) the type of action space being solved, (2) the method used to handle safety constraints, and (3) whether an additional environment model is required. We have discussed these key differences between our algorithm and that of the work [1] in detail in both the introduction and related work sections of the manuscript.
>
> **Q2: By the way, apart from the hybrid setting, are there any other differences between the algorithm in this paper and the one in [1]? Intuitively, the theoretical part seems essentially the same. Considering that the hybrid setting doesn’t seem to introduce additional challenges for the theoretical analysis, am I misunderstanding something?**
>
> Thank you for pointing this out. The work [1] first integrates real-world data with a learned model to perform online learning of environment dynamics, and then applies PPO-Lagrangian on truncated predicted trajectories to update both the policy and the Lagrangian multiplier, thereby achieving policy optimization under safety constraints. While the work [1] uses a Lagrangian method for constrained policy optimization, our approach employs primal policy optimization to address the constrained policy optimization problem. Moreover, the work [1] does not provide a detailed theoretical analysis. There are notable differences between work [1] and our algorithm, including: (1) the type of action space being addressed; (2) the method used to handle safety constraints; and (3) whether an additional environment model is required.
>
> In our study, the hybrid-action outputs satisfy the convergence properties of both the discrete action and continuous action value estimations as well as the policy networks. Therefore, building on the theoretical foundations established in prior research, we extend the action space and derive the convergence properties of our algorithm. Our theoretical analysis primarily focuses on the convergence characteristics of the safety constrained problem after the action space expansion, and the derivations in our work are centered on this aspect in detail.
>
> **Reference**
>
> [1] Jayant, A. K., & Bhatnagar, S. (2022). Model-based safe deep reinforcement learning via a constrained proximal policy optimization algorithm. Advances in Neural Information Processing Systems, 35, 24432-24445.
>
> [2] Hausknecht, M., & Stone, P. (2015). Deep reinforcement learning in parameterized action space. arXiv preprint arXiv:1511.04143.

---

> > ### Comment · Reviewer_33v4 · 2025-08-09
> >
> > Thanks for the response. Most of concerns are addressed and I have raised my score.

---

### Official Review · Reviewer_iyh9 · 2025-07-02

**Clarity:** 2
**Significance:** 2
**Originality:** 3
**Rating:** 4
**Confidence:** 3

**Summary:**

This paper addresses the problem of RL in environments with hybrid action spaces under safety constraints. In real-world applications like robotics and autonomous driving, traditional hybrid-action RL methods focus mainly on maximizing reward and often fail to ensure safety requirements in complex parameterized action spaces. The authors try to solve the problem by incorporating the safety constraints and using online RL to solve the constrained optimization problem.

**Questions:**

None

**Ethical Concerns:**

["NO or VERY MINOR ethics concerns only"]

**Final Justification:**

The authors responses have addressed many of my concerns, so I raise my score  to 4.

**Limitations:**

Yes

**Quality:**

3

**Strengths And Weaknesses:**

# Strength
- The authors formalize the Constrained Parameterized-action Markov Decision Process, which models RL tasks involving both hybrid action spaces and cost constraints. The idea is novel.

- The authors propose Constrained Hybrid-Action Policy Optimization, a new RL algorithm that can optimize policies in hybrid action spaces while strictly enforcing safety constraints. The proposed algorithm adopts a dual actor-critic architecture, and uses a direct policy gradient approach to avoid unstable Lagrange multiplier tuning.

- This paper provides theoretical guarantees for the convergence and safety of CHPO, showing that it learns policies that maximize reward without violating the cost constraints. Experiments on four tasks show that CHPO outperforms existing baselines in balancing reward maximization and safety costs.

# Major weaknesses
- Although online RL under safety constraints can solve some problems, the probability of risk occurrence at the initial state cannot be avoided, which is important in the real world since some occurrences of risk may make the training process stop. Moreover, as Figures 2, 4 and 5 show, the costs of CHPO are larger than the baselines and limitations for a long iteration, which may be intolerable in the real world. The usage in the real world may be limited. In my opinion, offline RL algorithms can achieve good performance in the tasks with safety constraints. From this perspective, the novelty of the motivation may be limited.

- The results in Table 1 show that CHPO can maximize the average reward under the safety constraints, but the variance may be too large, leading to the costs surpassing the limitations. As Figure 2 shows, CHPO can achieve much larger rewards when the costs surpass the limitation, which means the results may not be strong enough to confirm the theory of the authors.

- The baselines are out of date, with the most recent year being 2019. Is there any new method?

# Minor weaknesses
- The computation of the reward advantage function in Line 190 can be described in the appendix.

---

> ### Author Rebuttal · Authors · 2025-07-28
>
> We appreciate the time and effort you are dedicated to providing feedback on our paper and are grateful for the meaningful comments.
>
> **Q1: The probability of risk occurrence at the initial state cannot be avoided.**
>
> Thank you for bringing our attention to the offline setting. You raise a highly valuable and thought-provoking question. Safety constraints in hybrid action spaces are also in high demand in real-world applications, such as in gaming[1][2] and in recent IL + Online RL studies[3][4]. Our work simultaneously addresses the problem of safe policy optimization in hybrid action spaces, providing both a theoretical foundation and technical support for future research on constrained hybrid-action policy optimization in offline settings. Moreover, real-world data is used to build high-fidelity scenarios through world models, providing a feasible paradigm for online closed-loop interactions in safe RL. This makes it possible to sample from real-world environments with high safety requirements.
>
> The relatively high cost of CHPO in a long iteration is intentional to ensure global optimality. We incorporate the C/A ratio during training to ensure that the cost decreases gradually, allowing the algorithm to explore and achieve the maximum reward while satisfying the safety constraints. Furthermore, if higher policy safety is desired, we adjust the hyperparameters to balance the trade-off between reward and safety constraints, thereby ensuring strict adherence to safety constraints.
>
> **Q2: The variance may be too large.**
>
> Thanks for the kind suggestion. Although the theory guarantees the safety of the policy, in practical implementations the bias in the value estimation of the cost value network occasionally cause the policy to exceed the cost threshold. However, if strict adherence to the cost constraint is required, our algorithm achieves this by tuning the hyperparameters to ensure that the final policy strictly satisfies the cost constraint. Furthermore, to enhance the rigor of our statements, we have adopted your suggestion and revised the manuscript to clarify that “CHPO maximizes the average reward while ensuring that the average cost across multiple seeds satisfies the safety constraint.”
>
> **Q3: The baselines are out of date.**
>
> Thank you for your thoughtful review. The baselines we use are classical approaches, which have also been referenced by other recent studies [5][6]. Since there is no existing algorithm specifically for constrained hybrid-action RL, we combine well-established hybrid-action RL methods with stable constrained RL algorithms.
>
> **Q4: The computation of the reward advantage function in Line 190 can be described in the appendix.**
>
> We appreciate your careful reading. We provide the computation of the reward advantage function in the appendix: $\hat{A}^{r}_t=-V^{r}(s_t)+r_t+\gamma  r\_{t+1}+\cdots+{\gamma}^{T-t-1}r\_{T-1}+{\gamma}^{T-t}V^{r}(s_T)$.
>
> **Reference**
>
> [1] Hausknecht, M., & Stone, P. (2015). Deep reinforcement learning in parameterized action space. arXiv preprint arXiv:1511.04143.
>
> [2] Fan, Z., Su, R., Zhang, W., & Yu, Y. (2019). Hybrid actor-critic reinforcement learning in parameterized action space. arXiv preprint arXiv:1903.01344.
>
> [3] Gao, D., Wang, H., Zhou, H., Ammar, N., Mishra, S., Moradipari, A., ... & Zhang, J. (2025). IN-RIL: Interleaved Reinforcement and Imitation Learning for Policy Fine-Tuning. arXiv preprint arXiv:2505.10442.
>
> [4] Albaba, M., Christen, S., Langarek, T., Gebhardt, C., Hilliges, O., & Black, M. J. (2024). Rile: Reinforced imitation learning. arXiv preprint arXiv:2406.08472.
>
> [5] Niu, Y., Xu, J., Pu, Y., Nie, Y., Zhang, J., Hu, S., ... & Liu, Y. (2021). Di-engine: A universal ai system/engine for decision intelligence.
>
> [6] Zhang, R., Fu, H., Miao, Y., & Konidaris, G. (2024). Model-based reinforcement learning for parameterized action spaces. arXiv preprint arXiv:2404.03037.

---

> > ### Comment · Reviewer_iyh9 · 2025-08-05
> >
> > I appreciate the authors' responses, which answer my questions on some points. However, some concerns still remain.
> >
> > - Although the authors claim that the performance is achieved under some constraint, such as tuning the hyperparameters, the extra experiment results seem missing. And the costs should not exceed the established limitations in some tasks.
> >
> > - In addition, the authors state that "the relatively high cost of CHPO in a long iteration is intentional to ensure global optimality". I am wondering whether the performance of CHPO remains effective at a relatively low cost, similar to other algorithms.

---

> > > ### Author Response · Authors · 2025-08-05
> > >
> > > Thank you for taking the time to re-evaluate our work and for your thoughtful follow-up. We hope the clarifications below address your concerns.
> > >
> > > **Q1: Although the authors claim that the performance is achieved under some constraint, such as tuning the hyperparameters, the extra experiment results seem missing. And the costs should not exceed the established limitations in some tasks.**
> > >
> > > Thank you for your response. We leverage the C/A ratio to ensure that the algorithm explores and achieves the highest reward while satisfying the safety constraints. We define C/A ratio $= \frac{\sum_{i}^{m}|\mathcal{N}{c_i}|}{\sum_{i}^{m}|\mathcal{N}_{c_i}| + |\mathcal{N}_r|}$ as the proportion of update steps dedicated to cost minimization relative to the total number of policy updates. By tuning the C/A ratio, we adjust the safety level of the policy to ensure strict satisfaction of the safety constraints. We conducted experiments in the *Moving* environment using 3 random seeds under different C/A ratios, demonstrating that our algorithm satisfies the established limitations through appropriate hyperparameter tuning. The results for the cost limit $\bar{c}_i=1$ are as follows:
> > >
> > > | C/A ratio |    1/10   |    1/6    |    1/4    |
> > > |:---------:|:---------:|:---------:|:---------:|
> > > |   Reward  | 1.41±0.55 | 1.44±0.48 | 0.98±0.54 |
> > > |    Cost   | 0.22±0.99 | 0.12±0.56 | 0.00±0.00 |
> > >
> > > From the results above, we observe that as the C/A ratio increases, CHPO satisfies the established limitations. When the C/A ratio is set to 1/4, CHPO achieves fully safe behavior.
> > >
> > > **Q2: In addition, the authors state that "the relatively high cost of CHPO in a long iteration is intentional to ensure global optimality". I am wondering whether the performance of CHPO remains effective at a relatively low cost, similar to other algorithms.**
> > >
> > > Thanks for the comment. The proposed primal policy optimization algorithm determines the gradient direction based on the relationship between the cost and the cost limit. It performs policy updates directly from the advantage function, without introducing additional hyperparameters or instability. This design helps prevent the policy from falling into poor local optima and ensures reliable performance. In contrast, other baseline algorithms that rely on Lagrangian multipliers are sensitive to their initialization and suffer from training instability, often resulting in suboptimal safety or reward performance. As shown in the table in Q1, when the C/A ratio is set to 1/6, CHPO achieves a higher reward than PADDPG-Lag (which reaches 1.30) in the *Moving* environment, while satisfying the cost limit $\bar{c}_i = 1$ and maintaining a relatively low cost overall.
> > >
> > > Thank you for your detailed and thoughtful feedback. We would gladly offer further clarification if it could assist in your final assessment.

---

> > > > ### Comment · Reviewer_iyh9 · 2025-08-06
> > > >
> > > > Thanks for your responses, many of my concerns have been addressed now.

---

### Note · Authors · 2025-08-13

We thank the AC and reviewers for the time and effort they have dedicated to our paper. Their feedback has been invaluable in improving the quality of our work, and we sincerely appreciate their professional, patient, and fair comments and suggestions. During the discussion phase, we had constructive exchanges: Reviewer iyh9 noted that "many of my concerns have been addressed now"; Reviewer 33v4 stated that "most of concerns are addressed" and increased the score; Reviewer jBVV mentioned that "these responses address most of my original concerns, though some theoretical details remain unclear," and maintained the score, expressing the hope that we integrate CHPO into the DLPA framework; Reviewer HetW indicated that "I have no other questions and will decide whether to increase the score after careful consideration." We are very grateful for these constructive comments, and we have improved the quality of this work in line with their suggestions.

---

### Decision · Program_Chairs · 2025-09-17

**Decision:**

Accept (poster)

**Comment:**

The paper proposes Constrained Hybrid-action Policy Optimization (CHPO), addressing reinforcement learning with hybrid discrete–continuous actions under safety constraints by formulating the CPMDP framework. The strengths include good motivation, strong theoretical guarantees, and convincing experimental results showing improved reward–safety trade-offs compared to baselines. Weaknesses lie mainly in a little limited environments and reliance on stylized assumptions in the proofs, but the authors provided clarifications, ablations, and additional results during the rebuttal that mitigated these concerns. The reviewers acknowledged that most of their concerns were resolved, and some raised their scores. Overall, the novelty, analytical results, and impact justify acceptance.